# Beijing's Selected Older Neighborhoods Measurement from the Perspective of Aging

**Cao Gan** [1], **Mingyu Chen** [1,*] and **Peter Rowe** [2]

[1]  School of Architecture, Tsinghua University, Beijing 100086, China; ganc18@mails.tsinghua.edu.cn
[2]  Graduate School of Design, Harvard University, Cambridge, MA 02114, USA; prowe@gsd.harvard.edu
*  Correspondence: cmy15@mails.tsinghua.edu.cn

**Abstract:** The older neighborhoods in Chinese cities are the main areas in which the elderly live. Both housing and the older adults have experienced the resonance process of co-growth and co-aging. Along with the acceleration of the aging population, the older neighborhoods with the aging physical space environment are increasingly unable to meet the needs of the elderly, which affects their health and well-being. Although the Chinese government has launched a program of retrofitting older neighborhoods to make them more hospitable for older residents through, for example, elevator installation and infrastructure improvement, these practices are merely associated with various neighborhood features. Most existing research starts from small-scale case studies and consequently lacks macro-level analysis of the characteristics of the physical spaces and aging population in older neighborhoods. Therefore, this paper selects old neighborhoods constructed from 1949–1999 in the central city of Beijing (within the Fifth Ring Road) as research subjects. The research begins with an analysis of the construction and evolution of the standards of selected older neighborhoods in Beijing and establishes measurement metrics for the spatial characteristics of both the neighborhood and aging population. This article concludes that older neighborhoods in central Beijing can be classified into seven clusters based on their spatial characteristics and three clusters based on aging population characteristics through K-means classification. Additionally, this paper conducts an overlay analysis of these two classification results to identify different spatial features of older neighborhoods within varying characteristics of the aging population and proposes suggestions for the renovation of selected old neighborhoods. The study aims to provide a reference for retrofitting older neighborhoods with the goal of creating an aging-friendly community, and to supply a scientific basis for empirical research on the middle scale and micro scale.

**Keywords:** aging; older neighborhoods; urban design measurement; neighborhood retrofitting

## 1. Introduction

Global aging is one of the essential development tendencies and has a profound impact on the socio-economic development of countries all over the world. Since China initially entered the "aging society" classification in the early twenty-first century, the degree of the aged phenomenon has continued to accelerate and deepen. According to data from China's National Bureau of Statistics, people over 65 accounted for 11.9% of China's population at the end of 2018. The World Population Prospects 2019 released by the UN predicts that the degree of aging in China will continue to rise in the future, reaching 26.1% in 2050, which is comparable to that of developed countries [1]. Facing the urgent pressure of providing for elderly people, China proposed a "90,7,3" pension policy (90% of the elderly care at home, 7% in the community, and 3% in institutions) during the eleventh Five-Year Plan (2006–2010). Living in a familiar environment has always been a common choice and need for elderly people [2,3], so "aging in place" has gradually become a consensus strategy for

major developed countries and international organizations since the 1960s [4]. However, research studies report that remaining in one's home and the community does not always lead to aging well, which specifically suggests an increased likelihood for poor physical and mental health among older adults living in neighborhoods characterized by the high residential stability of older residents [5,6]. These neighborhoods are characterized by years of neglect and disinvestment, poorly maintained roads and sidewalks, few service providers, and few opportunities for social interaction caused by gentrification [7,8]. In that context, older adults who are unable to relocate due to economic and social resource constraints are "stuck in place" instead of experiencing real "age in place" [7], which has been defined as "the ability to live in one's own home and community safely, independently, and comfortably, regardless of age, income, or ability level" [9]. The phenomenon mentioned above, spending old age in an aged residential environment, is more obvious in Chinese cities because most of the senior citizens stably live in the older neighborhoods built more than 20 years ago [10–12]. As the aging problem was not taken into account in early construction, the decaying living environment gradually failed to meet the needs of older residents and formed a growing contradiction [13–15]. For instance, the older neighborhoods have become gradually disconnected and isolated from the renewal and development of the surrounding urban areas, with a lack of public space in the internal environment and the safety of travel affected by cars. The increasing proportion of the elderly population in China means that older neighborhoods will serve as the main carriers of "aging in place" at this stage and in the future [16,17], so it is urgent to take certain measures to improve that process.

　　Extending the life span of older neighborhoods, rather than demolishing and rebuilding, is a reasonable and feasible option. First of all, from the perspective of sustainable development, it is unrealistic to dismantle all of the neighborhoods, and retrofitting is more economical and environmentally friendly than rebuilding [10,16,18]. Besides, from the angle of architectural culture protection, older neighborhoods are an indispensable part of reading the overall built environment of the city, and creating a high-quality living space is in line with new international trends. The 2018 Davos Declaration highlighted the promotion of the concept of a high-quality Baukultur in Europe [19], where "Bau-" refers to the verb to build and "Kultur" refers to the noun for culture [20]. It means the building is culture and creates space for culture, so it encompasses not only cultural heritage but also all existing buildings, public spaces and contemporary creation, aiming to create cultural identity and diversity for the built environment we want to live in [21]. In addition, the older adults are, the more important it is for them to live in a familiar environment [7,22]. Compared to younger people, older adults may have stronger feelings of place attachment [23], which may positively affect health and well-being by helping individuals to develop and maintain a personal identity and a sense of belonging [24]. Taking older neighborhoods associated with danwei (or state-owned work unit) as an example, danwei communities were built to meet the housing needs of employees from government agencies, social service institutions or factories during China's planned-economy period (1949–1978) and the following two decades. They combined a working quarter and living quarter and were organized like a small society in the form of a gated and walled-off community providing fully equipped service infrastructure [25]. Although nowadays this work-unit system has been abandoned and the large numbers of working quarters have been regenerated, relocated or demolished, most of the living quarter has been left behind, with retired workers and their generations living inside. The social connection between older residents in the community is a dual relationship of "neighbors + colleagues", or even a triple relationship of "neighbors + colleagues + relatives" [26], and this strong social bond is beneficial to health and well-being [27,28]. Therefore, implementing the environmental retrofitting of older neighborhoods, rather than rebuilding, is better for the recognition of urban culture and the health of the elderly.

　　Shaping older neighborhoods with the goal of the aging-friendly community has been widely recognized. The concept of the aging-friendly community is based on the ecological model of aging from environmental gerontology, which proposes that the health and well-being of the elderly are a result of the interaction between the competence of the individual and the environmental pressure of the context,

emphasizing the maintenance of a better "person-environment fit" in the dynamic aging process [29]. Based on this, the active aging theoretical framework was constructed and applied as the action purpose of an aging-friendly city and community, creating a series of evaluation index systems [30,31]. Scharlach (2015) further suggested that an aging-friendly community includes five characteristics: livability, elder friendliness, lifespan development, communality and transactionality [7]. While there is little research in the field of elderly-friendly environments in China [32], most of the studies focus on the empirical observation of community cases and the investigation of the willingness of living based on the theory of the older individuals' lifestyle and the hierarchy of needs [33]. Specifically, current research give design solutions and policy recommendations by recording and discovering the elderly's behavioral needs and problems when they are using indoor spaces, community public spaces or service facilities. However, in terms of the scale and urgency of retrofitting of older neighborhoods in China, they cannot be dealt with individually. Thus, it is necessary to construct the older neighborhoods measurement system at the city level to clarify the goal and content of environmental retrofitting [10].

Aging friendliness emerges in the interaction between individuals' characteristics (e.g., health, social and economic resources) and the environmental context (e.g., the physical and social infrastructure) [7], so this study aims to construct a measurement index system for overall older neighborhoods on the macro scale, based on the two content dimensions of aging friendliness, which are aging characteristic metrics and physical space characteristic metrics. For the former, current research mainly uses the following variable to analyze the features of the elderly population: age group (65–70, 71–79, 80+ years); marital status (unmarried, married); educational level (high school or below, college or above); revenue level (below 3000, above 3000); revenue sources (pension dependence, relief dependence, family support dependence); and activities of daily living, abbreviated as ADL (complete self-care ability, impaired self-care ability, inability of self-care) [34–36]. For the latter, physical neighborhood characteristics that have been investigated in previous studies can be grouped into two categories: neighborhood social attributes and built attributes [34]. In terms of features of the built environment, current research mainly focuses on the following design features that are important for sustainability and active travel: (1) land use density and morphology, measured as floor area ratio (FAR), building density, average building height and building typology; (2) mixed land use, measured as a Simpson index based on the floor areas of different land uses; (3) public greenspace and landscape, measured as green ratio; (4) elderly-oriented facilities, measured as elevator installation; (5) community governance and operation, measured as property fees [37–41]. Besides, institutional factors (developed by work units, government, and private sectors) that are unique to China are also important because of the transition from a planned to a market economy [42]. Therefore, based on a literature review, this paper firstly intends to build two metric measurement systems for older adults and the physical environment. Then, through a cluster analysis and overlay analysis of the two index systems, the specific characteristics of community and people in the process of "synchronous aging" can be supplemented as the scientific basis for empirical research at the middle level and micro level, further promoting the effectiveness of environmental retrofitting practices in older neighborhoods.

## 2. Materials and Methods

### 2.1. Research Object

The study chose Beijing, the capital of China, as its city case study since it is a representative city with rapid urban development and all the major types of older neighborhood [42]. Beijing provides a typical case to study both the institutional and physical environmental factors of neighborhoods. Besides, according to the Sixth census published in 2010, Beijing is one of the cities with the highest proportion of elderly people in China [43]. It is worth mentioning here that "older neighborhoods" in this article refers to those that were built more than 20 years ago and have a high residential stability of older residents. Although many hutong courtyards located in the old city of Beijing (within the Second Ring Road) also belong to older neighborhoods and have serious aging problems, this paper

focuses on residential quarters built after the founding of the People's Republic of China in 1949 as the "selected older neighborhoods". Since there are obvious differences in the physical spaces and retrofitting models, they are not suitable to measure together. Besides, unlike the gated communities, there is already a consensus on the generally poor physical environment quality and population of older residents with lower socioeconomic status in hutong courtyards, and these therefore do not need to be evaluated [44]. The research scope of this article is limited to the central city of Beijing (within the Fifth Ring Road), which includes the whole area of Xicheng District, Dongcheng District and parts of Haidian District, Chaoyang District, Fengtai District, Shijingshan District, and Daxing District. This research area covers an area of 668.62 km$^2$, including a total of 113 sub-districts. The reason why this area was selected was that it has been the primary zone of urban development and construction after the founding of New China in 1949. In addition, this area not only has the most concentrated zone of older neighborhoods but is also the area where the problem of aging urban residents is the most severe [11]. Through collating data from real estate transaction websites (including HomeLink, Anjuke, and 58.com), this research found that after 1949 there were a total of 2460 older neighborhoods in Beijing, and 2246 older neighborhoods within the Fifth Ring Road of Beijing (over 90% of the total).

### 2.2. Qualitative Method

This article reviewed the literature of two journals called the *City Planning Review* and the *Architectural Journal* and extracted the contents related to old neighborhoods from the following aspects: national policy system and background, residential planning structure and layout and relevant design standards (average number of residents per household, per capita living area, per household living area, per capita public green area and residential density). Furthermore, this study conducted a qualitative description of the development process of selected old neighborhoods in Beijing in stages. Further, this paper quantified the magnitude of urban expansion and analyzed the expansion types and spatial distribution patterns of neighborhood construction in different stages. Outlying, edge-expansion, and infilling were three types frequently used to describe urban growth [45]. Urban growth types were defined by the landscape expansion index (LEI):

$$\text{LEI} = \{S_v / (S_O + S_V)\} \times 100 \tag{1}$$

where $S_O$ is the intersection between the buffer zone and the occupied category for a newly constructed residential neighborhood, $S_V$ is the intersection between the buffer zone and the unbuilt area in the new neighborhood. Urban growth type is defined as outlying, infilling, and edge-expansion when LEI = 0, LEI > 50, and0 < LEI ≤ 50, respectively. A smaller value for the buffer distance indicates a more stable LEI value; thus, the buffer distance was set to 1 m in this study [45].

### 2.3. Quantitative Method

The study built a metric measurement system of physical environmental characteristics of selected old neighborhoods from the following three aspects: institutional characteristics, morphological characteristics, and construction and operation quality. The selection of indicators is shown in Figure 1. The data included network data of old neighborhoods constructed from 1949–2000, excluding abnormal data such as villa settlements (construction time, property fee, property right, elevator installation, etc.); the data of the elderly population in the sub-districts of Beijing; land use map of Beijing; building data and neighborhood boundary data for Beijing; and Sentinel-2 satellite data for Beijing. Three methods were adopted in the quantitative analysis process, as shown in Figure 1. The first was to directly process the GIS data, that is, to calculate or assign data within the boundary of old neighborhoods (building density, plot ratio, average building layers, building typology, and green ratio) and the degree of land use mix within the 300 m buffer zone of the old neighborhoods on GIS platform [46]. The second was to capture data from websites such as HomeLink, Anjuke, etc. and input data into a GIS platform

(construction time, property fee, elevator installation, etc.). The third was to collect relevant data and manually input it into a GIS platform (associated with danwei community, property rights etc.).

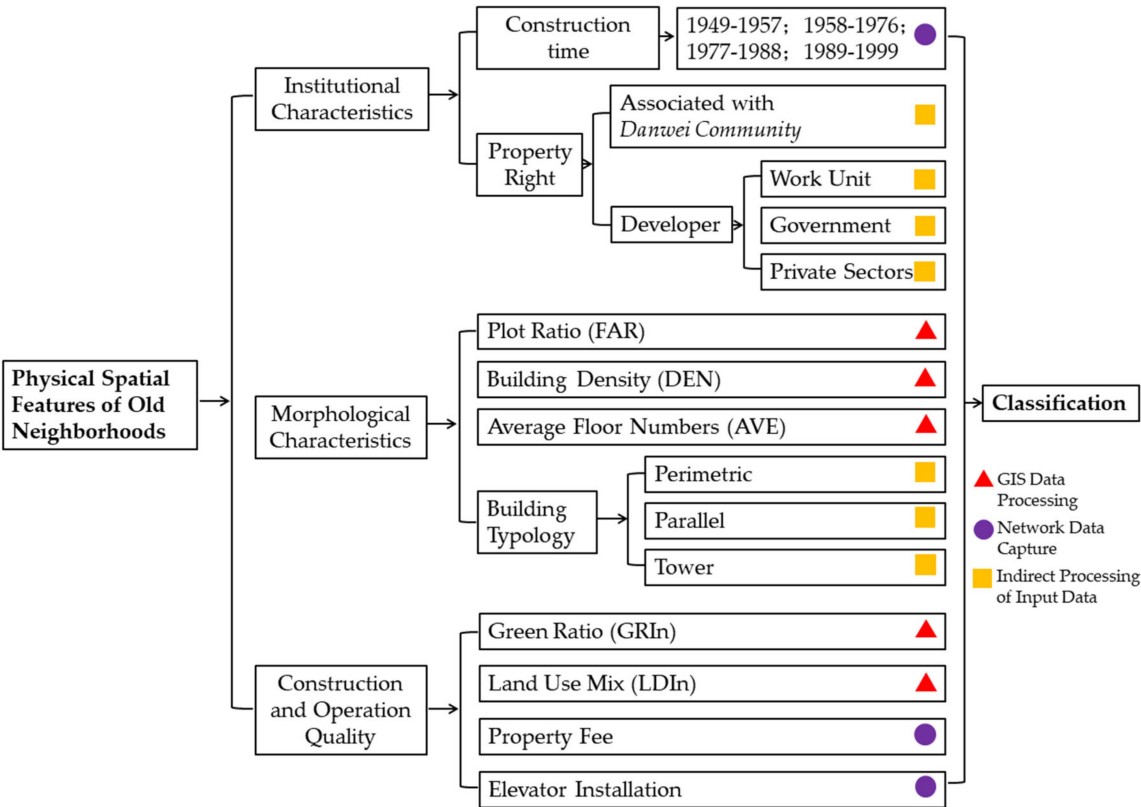

**Figure 1.** The measurement metrics and quantitative methods for the physical spatial characteristics of selected older neighborhoods in Beijing.

Calculation methods for the specific indicators were as follows:

(1)　Construction Time Index:

On the basis of the data of older neighborhoods captured from HomeLink, the research divided construction time into four periods for measurement through policy system analysis in a qualitative method: a construction time index was assigned to 1,2,3,4 when the older neighborhoods were constructed from 1949 to 1957, from 1958 to 1976, from 1977 to 1988, and from 1989 to 1999, respectively.

(2)　Property Right Index:

In Beijing, political economic and social institutions changed dramatically from 1949 to 1990 and the property right features of the selected older neighborhoods are various. So, this study collected information on current older unit neighborhood developers from websites such as 58.com, Anjuke, HomeLink and other online fora and assigned values: the property right index was assigned to 1, 2, and 3 when the developer of older neighborhoods were the work, government, and private sectors, respectively. Additionally, the paper collected information and gave identification on whether the older neighborhoods were associated with a danwei community (see the definition in the Introduction) since these once played a significant role in Chinese cities and can be regarded as a particular type of older neighborhoods: this index was assigned to 1, 0 when the older neighborhoods were associated with danwei community or not, respectively.

(3)　Plot ratio, building density and average floor numbers within the boundary of older neighborhoods:

Plot ratio, building density and average floor numbers reflect morphological characteristics and organization patterns of older neighborhoods. This article calculated plot ratio, building density and

average floor numbers within the boundary of each older neighborhoods through the intersect tool in the ArcGIS platform:

$$FAR = (\sum S_i \times N_i)/S_O \tag{2}$$

$$DEN = \sum S_i/S_O \tag{3}$$

$$AVE = \sum N_i/n \tag{4}$$

where $S_i$ is the area of buildings in older neighborhoods, $N_i$ is the number of buildings in older neighborhoods, $S_O$ is the area of older neighborhoods, and FAR, DEN and AVE range from 0 to 1.

(4) Building Typology Index of Older neighborhoods

Through analysis of literature, this paper divided the building layout of older neighborhoods into three types: perimetric, parallel and tower [47,48], which are shown in Table 1.

**Table 1.** Building layout pattern of residential neighborhood.

| Typology | Definition | Layout Diagram |
|---|---|---|
| Parallel | "Parallel layout" refers to the combination form of buildings arranged in rows according to specific directions and reasonable distance specified by sunshine spacing. Buildings are mostly arranged in row townhouses or strip units. | |
| Perimetric | Buildings in residential neighborhood with the "perimetric layout' are arranged around the neighborhood or courtyard. This kind of arrangement tends to form a closed or semi-closed, relatively independent and complete inner courtyard. | |
| Tower | "Tower layout" includes low-rise single courtyard houses, multi-story point houses and high-rise towers. | |

If "parallel layout" appeared in the neighborhood then the "parallel layout index" equaled 1; if not, then 0. If "perimetric layout" appeared in the neighborhood then the "perimetric layout index" equaled 1; if not, then 0. If "tower layout" appeared in the neighborhood then the "tower layout index" equaled 1; if not, then 0.

(5) Green ratio within older neighborhoods

The research selected Senitel-2 remote sensing data to calculate the Normalized Difference Vegetation Index (NDVI) through the Raster Calculator on the ArcGIS platform and it extracted raster with NDVI ≥ 0.3 as the scope of green space [49]:

$$NVDI = (NIR - R)/(NIR + R) \tag{5}$$

where NIR is the Near Infrared Band and R is the Red Band. Besides, this study calculated the green ratio of each older neighborhoods through means of Intersect Tool in the ArcGIS platform:

$$GRIn = S_G/S_O \tag{6}$$

where $S_G$ is the area of green space within the boundary of older neighborhoods and $S_O$ is the area of older neighborhoods.

(6) Land Use Mix Index within older neighborhoods

The Land Use Mix Index reflected the functional diversity and uniformity of the surrounding environment of residential neighborhoods. This paper calculated the Land Use Mix Index within

the 300 m buffer zone of older neighborhoods based on the Simpson Diversity Index method in statistics [50]:

$$\text{LDIn} = 1 - \sum_{i=1}^{n} (S_i / S_n)^2 \tag{7}$$

where $n$ is the number of different types of land use within the buffer zone, $S_n$ is the total area of all land uses within the buffer zone, $S_i$ is the area of Type i land use, and LDIn ranges from 0 to 1.

In addition, the article selected analytical data of the aging population investigation in Beijing to analyze the aging conditions of different older neighborhoods. It conducted quantitative analysis from two aspects, the proportion of elderly people and elderly population characteristics, and furtherly selected 16 indicators. The selection of indicators is shown in Figure 2.

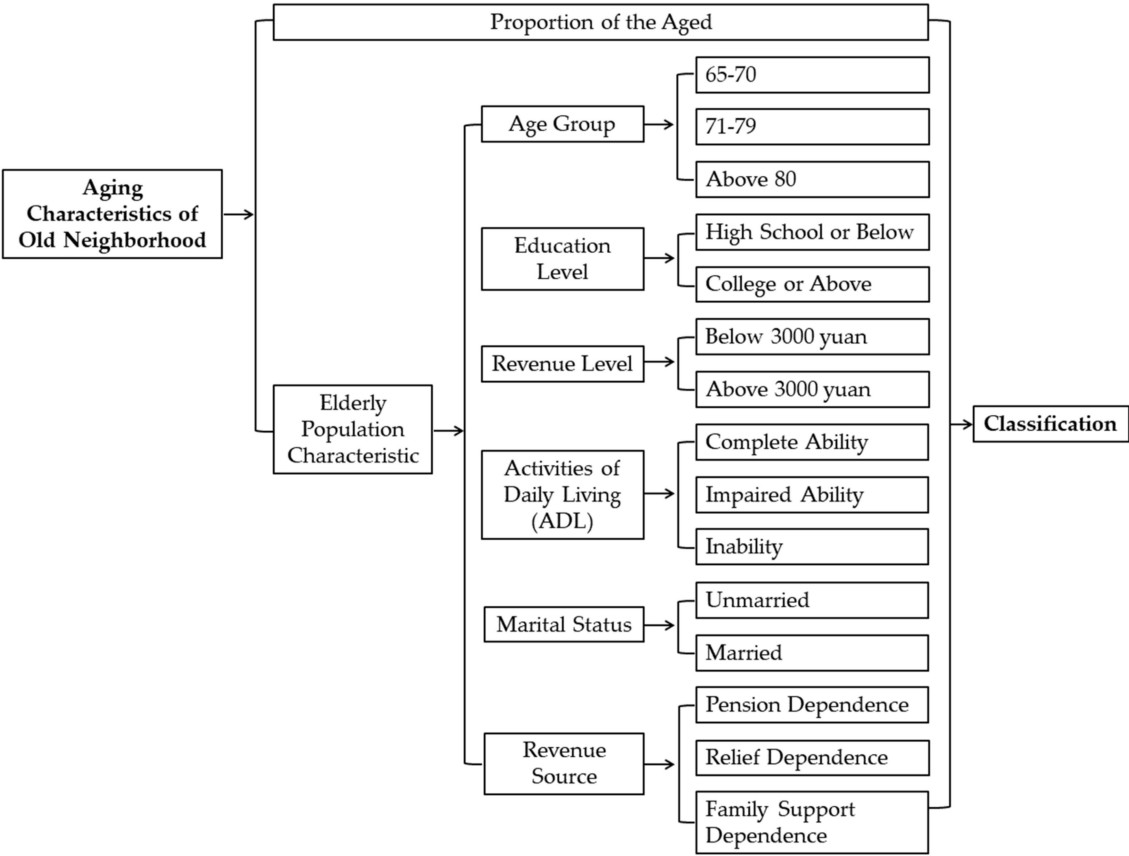

**Figure 2.** The measurement metrics and quantitative methods for the aging characteristics of the selected older neighborhoods in Beijing.

This paper then made a classification analysis of both the physical environmental features and the elderly population features of the selected older neighborhoods, which were universally applied to current research on the housing choice behavior of older neighborhoods [38]. There were mainly three clustering methods in the classification analysis: K-means clustering, hierarchical clustering and two-step clustering [51–53]. The comparison among these three methods is shown in Table 2.

Compared to the two other algorithms, the K-mean clustering algorithm is efficient and easy to implement and is universally used for the analysis of large samples [53]. The sample size of this study was large and there were little solitude samples in this research. Therefore, this study adopted the K-means classification algorithm to conduct large sample classification analyses on physical environmental characteristics and aging characteristics of older neighborhoods and selected the Sum of Squared Error (SSE) to check the results of the classification and determine the optimal number of clusters [52]. Finally, this paper observed the relationship between clusters of physical environmental

features and aging features and analyzed the community selection preference characteristics of different types of elderly people.

**Table 2.** A comparison of different clustering algorithms.

| Clustering Algorithm | Principle | Advantages | Disadvantages |
|---|---|---|---|
| K-means Clustering | Based on distance similarity judgment. When the distance of two samples is small enough, they will be divided into the same cluster. | Easy to implement, efficient and fast, capable of clustering analysis of large sample. | Sensitive to data noise and solitude samples. |
| Hierarchical Clustering | First, calculate the distance among samples. Merge the closest data into a small cluster. Then calculate the distance among clusters and merge the nearest clusters into a larger one. | There are few restrictions. The algorithm does not need to set the number of clusters in advance and will find the hierarchical relationships between the clusters. | The algorithm is of high calculation complexity and sensitive to solitude samples. |
| Two-step Clustering | The algorithm consists of two stages. In the pre-clustering stage, the samples in a dense area are clustered into many small sub-clusters. Then, the sub-clusters are merged one by one until reaching the expected number of clusters through agglomerative clustering method. | The influence of exception values can be excluded and the results of the first step can provide certain reference value for the K value in the second step. | The algorithm is of high calculation complexity. |

## 3. Qualitative Description

### 3.1. The Four Stages of Residential Development

Based on the combination of research in Chinese urban development policies and a literature review, this study summarizes the development process of selected older neighborhoods in Beijing into four phrases: 1949–1957, 1958–1975, 1976–1987 and 1988–1999. After the founding of China in 1949, the government adopted policies of a planned economy and prioritized the development of heavy industry [54]. The housing redistribution system featured as "low rents, high subsidies, welfare systems, physical distribution" [55], which caused the construction of danwei communities. The neighborhood built at that time had insufficient living areas for individuals, with many families sharing rooms. The space design focused on using social interaction space to improve life satisfaction [27,56]. In terms of planning structure, the housing blocks with three to four floors and a perimetric layout were built mostly in the 1950s. After the second Five-Year Plan began in 1958, policies such as "Socialism Builds Better, Faster and More Economically" and "Rammed-Earth House" formulated under the influence of the left-leaning policy caused housing development to slow significantly. At this stage (1958–1975), Beijing completed an average of 700,000 square meters per year, which was far lower than the construction volume during the 1950s [57] and also resulted in the construction of more than 1 million square meters of new simple houses under the one-sided pursuit of a policy reducing residential buildings [58]. Most residential quarters had five to six floors and a parallel layout with a 7–8 ha floor area [59]. As the fifth Five-Year Plan began and the cultural revolution ended in 1976, the commercialization of housing started under the impact of China's reform and opening-up policy and experimentation with the private market, leading to significant changes in neighborhood construction.

For instance, with the completion of 36 high-rise residential buildings on the south side of Beijing Qiansanmen Street in 1976, the number of high-rise constructions with tower layout had gradually increased and the scale of residential construction also expanded to the level of residential districts with a floor area above 10 ha. Although Zhang Kaiji and other experts had continuously suggested that Beijing should strictly control high-rise construction and replace high-rises buildings with a "multi-layer and high-density neighborhood" to play a role in saving urban construction land, most of the residential development plans still accepted high levels and high plot ratio [60]. The last stage was from 1988 to 1999, because after 1988 market reform officially entered a comprehensive pilot phase and the real estate industry developed rapidly. The housing development and distribution system, which was based on an increasingly privatized market, was initially established until 1998, at which point the housing welfare system was formally canceled and the danwei community model was thoroughly abandoned [61]. Statistics show that Beijing's average housing completion area has exceeded 6 million square meters since the 1990s [62]. In summary, the development of Beijing's older neighborhoods can be divided into four stages with the time boundaries of 1958, 1976, and 1988.

*3.2. Residential Spatiotemporal Distribution*

This study also analyzes the growth characteristics of selected older neighborhoods in Beijing in four sections from the perspective of time and space, as Figure 3 shows below. About 5.17 million square meters of new residential districts were built from 1949 to 1957 and 90% of the total number were the outlying type outside the city center (see Section 2.2), mostly concentrated in the suburbs on the west side of the Second Ring Road (the city center was in the second road at that time). Since the renovation of Beijing old city was mentioned in the urban planning agenda during the founding of China in 1949, a small number of infilling neighborhoods were also distributed in the old city. Unlike in the first phase from 1949–1957, the outlying growth type accounted for the majority. The proportion of infilling type and the outlying type was 42% and 54%, respectively, in the second stage (1958–1975). About 9.92 million square meters of new neighborhood were built, mostly located inside and outside of the Second Ring Road and both sides of Qianmen Street. This kind of distribution occurred since a large number of bungalows were removed and new neighborhoods were built outside the Second Ring Road to solve the housing shortage and relocation plan. During the 1976–1987 stage, the construction area was about 30 million square meters and the proportion of infilling type (51%) exceeded that of outlying type (40%). The new neighborhoods were concentrated between the Second Ring Road and the Third Ring Road. At the same time, the spatial layout in the northwestern part of the city and the axis of Changan Street also appeared, with expansion to the fourth and fifth ring roads. This kind of distribution was partly due to the accelerated urbanization process after the reform and opening up, which brought a housing demand of at least 4.4 million square meters [59]. In addition, the direction of housing construction growth was also affected by the restoration of overall urban planning management. In the last stage (1988 to 1999), the completed neighborhoods exhibited a more obvious layout that broke the Third Ring Road and there were also multiple housing clusters separated from the main urban residential construction areas. The cluster formation was in line with the Beijing Urban Master Plan 1992, which proposed that the key urban construction should gradually shift from urban areas to the majority of suburbs and the scale of new cities should be expanded [63].

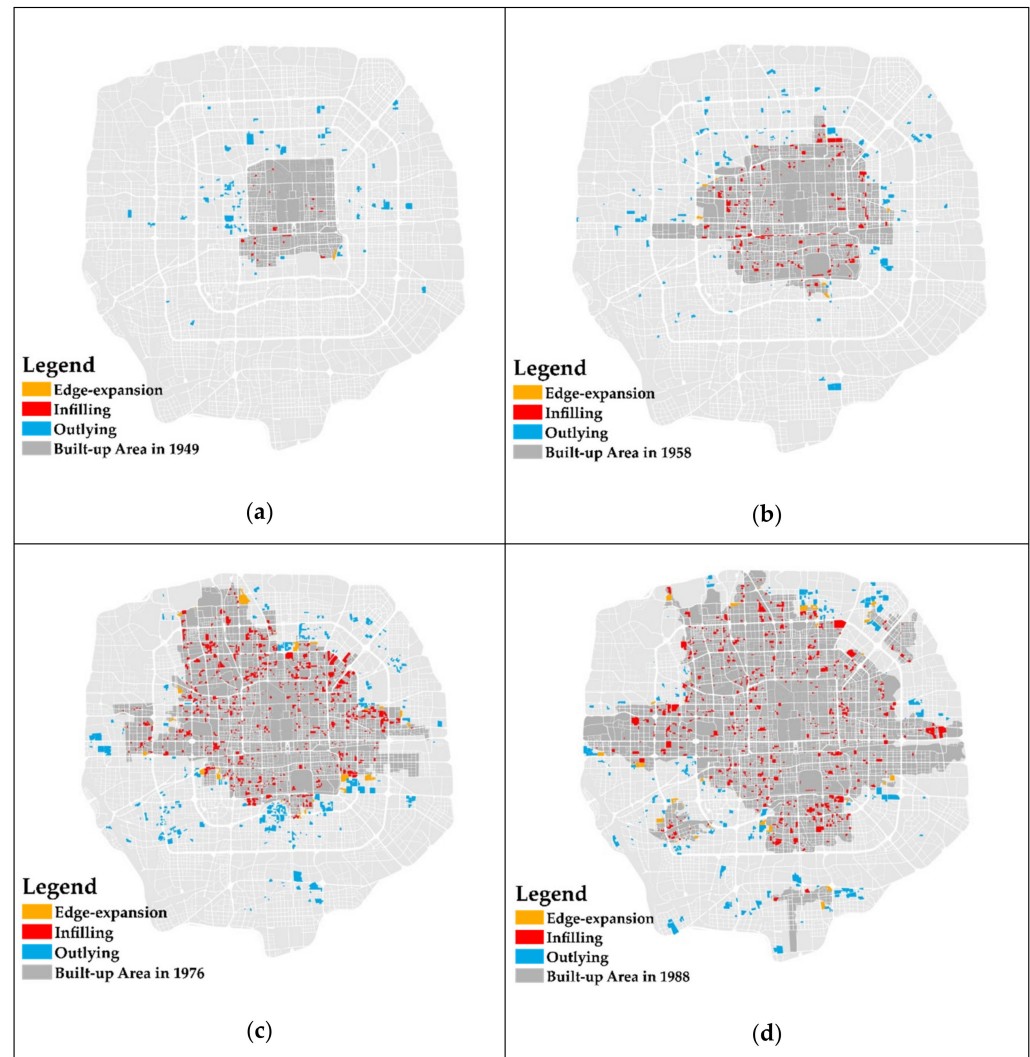

**Figure 3.** The spatiotemporal distribution of the four phases of the selected older neighborhoods: (**a**) 1949–1957; (**b**) 1958–1975; (**c**) 1976–1987; (**d**) 1988–1999.

## 4. Quantitative Results

### 4.1. Data Description

The results of the descriptive statistics of the physical environmental characteristics are shown in Figure 4 and Appendix A.

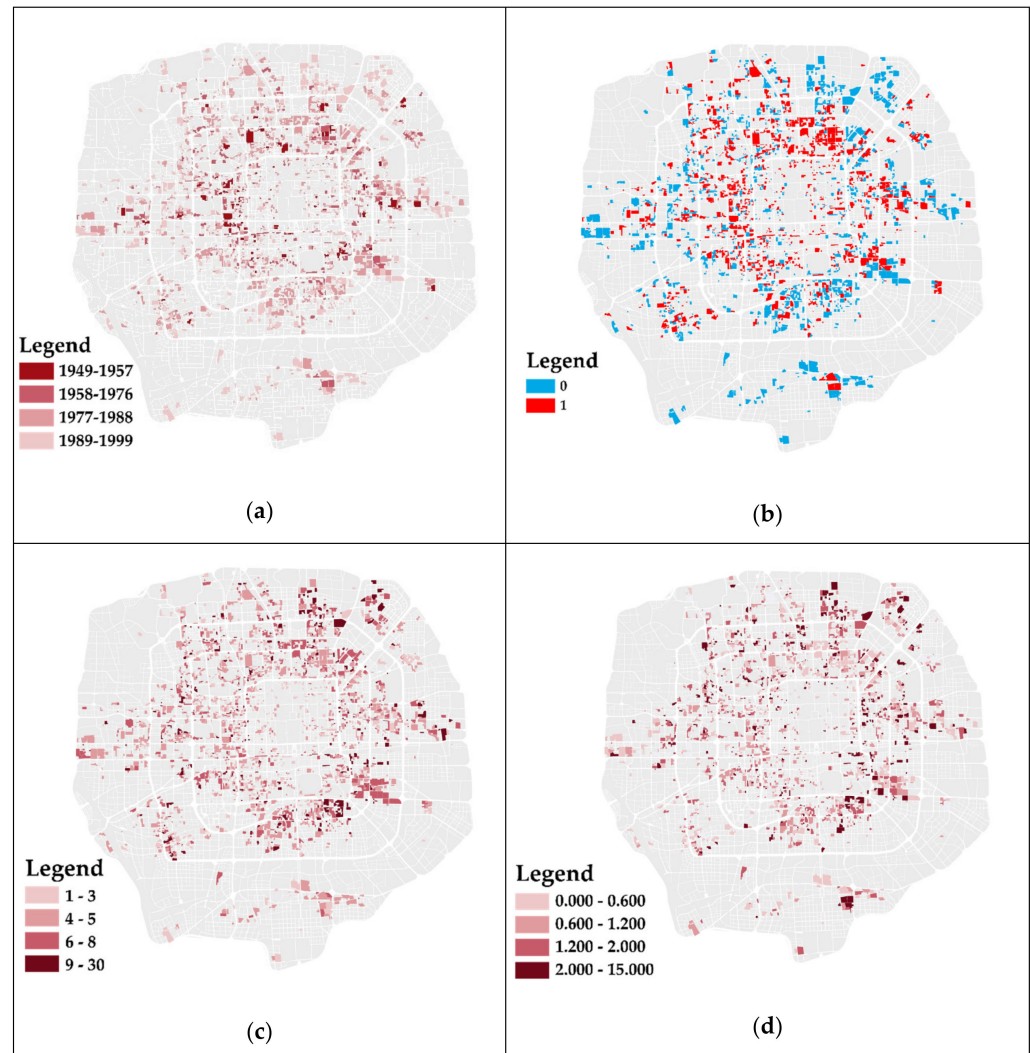

**Figure 4.** The spatial distribution of the physical spatial characteristics of selected older neighborhoods: (**a**) construction time; (**b**) associated with danwei; (**c**) average floor number; (**d**) property fee.

From the statistics, this research discovered that the majority of older neighborhoods in Beijing were constructed in the third stage (1977–1988) and the fourth stage (1989–1999). Additionally, the majority of the selected older neighborhoods in Beijing used to be danwei communities. The building arrangements of older neighborhoods are mainly the 'parallel layout' combined with a small proportion of 'perimetric layout' and 'tower layout'. The average floor numbers of the majority of the old residential neighborhoods in Beijing are less than six. Besides, the Land Use Mix Indexes of older neighborhoods are relatively high. However, their green ratio and property fee are relatively low and most of the older neighborhoods have not installed elevators.

The results of the descriptive statistics of the aging characteristics are shown in Figure 5 and Appendix B.

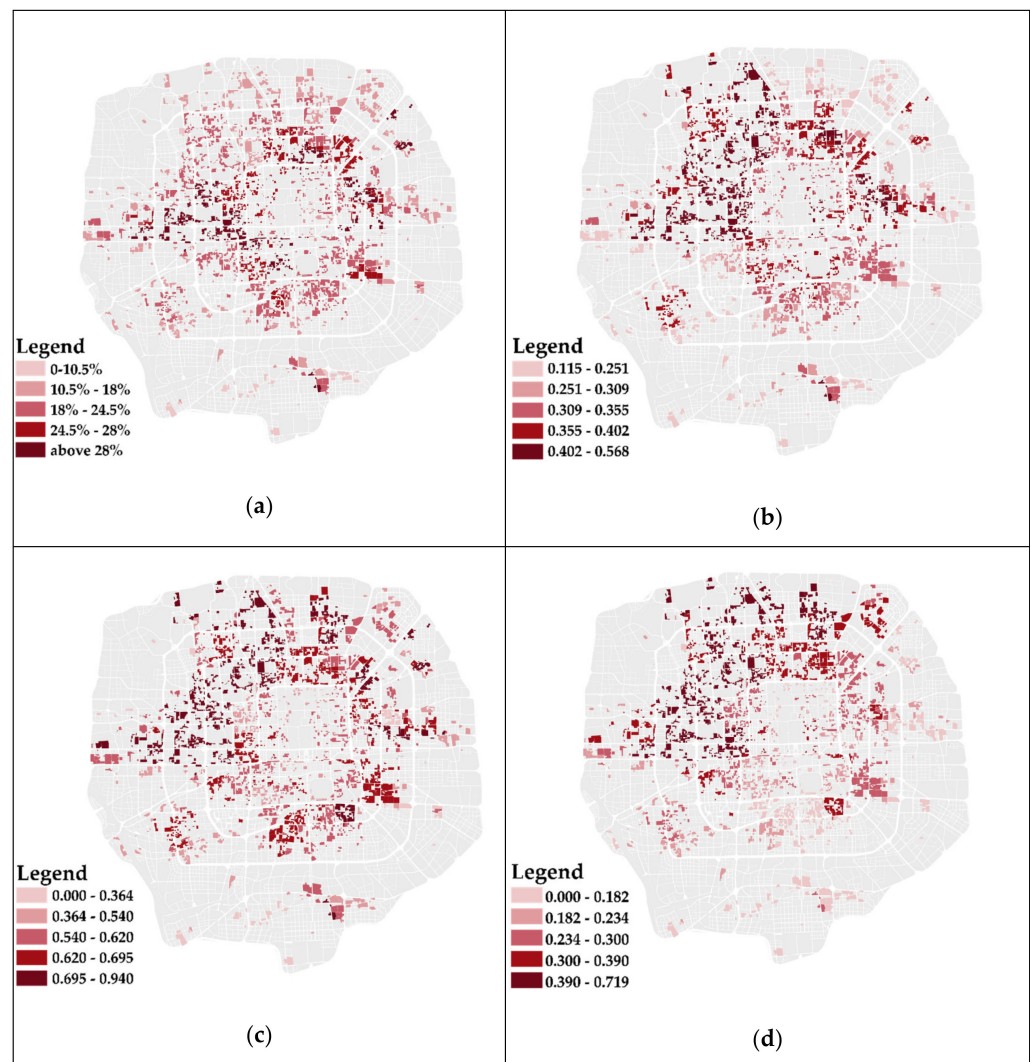

**Figure 5.** The spatial distribution of the aging characteristics of selected older neighborhoods: (**a**) proportion of elderly people; (**b**) age group: above 80; (**c**) revenue level: revenue above 3000 yuan; (**d**) educational level: college or above.

This paper discovered that the proportion of the aging population in older neighborhoods within the Fifth Ring Road in Beijing was relatively high. Most of the aging population in the selected older neighborhoods in Beijing was below 80 years old, had a high school education or below, and their monthly revenue generally exceeded 3000 yuan. Additionally, in the selected older neighborhoods, the proportion of married or completely self-care older people were relatively high. Different areas in the central city of Beijing also reflected different characteristics of aging: firstly, the older neighborhoods located in the west or the northeast of the central city were confronted with more severe aging problems; secondly, the older neighborhoods situated in the west or the northwest of the inner city have a high proportion of the "oldest old" (aged above 80 years old), and the aging population dependent on pensions and the unmarried elderly population were also relatively high; thirdly, the older neighborhoods in the south or southeast of the center demonstrated the main aging population characteristics of the 1/2 self-care level, high school education or below, and dependence on relief.

*4.2. Results of K-Means Classification*

This paper conducted a K-means classification of the physical environmental characteristics of older neighborhoods in Beijing and analyzed the SSE when K was from 4 to 10 (Figure 6).

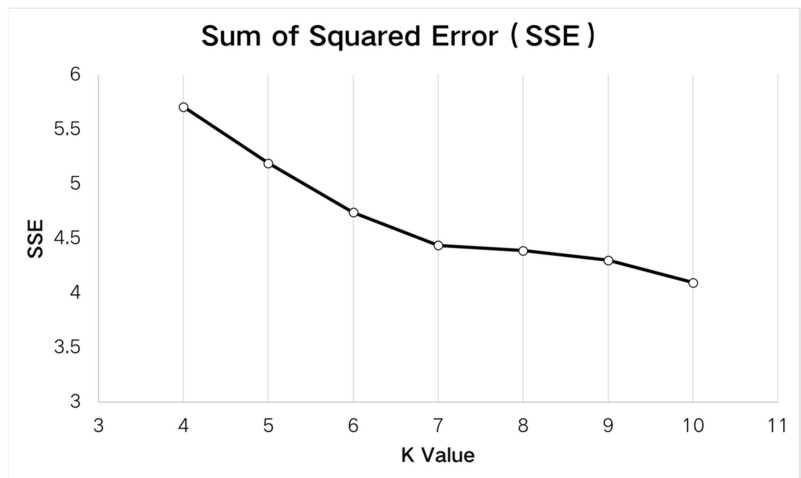

**Figure 6.** The SSE result analysis of K-means classification of physical environmental characteristics of selected older neighborhoods in Beijing.

Through the SSE analysis, this study discovered that SSE decreases with the rise of the K value. However, when K ≥ 7, SSE decreased much more slowly with the rise in K value, which meant that the classification effect did not enhance regardless of the increase in the number of clusters. Therefore, this article determined the optimal number of groups as seven. It conducted K-means classification according to seven categories and the results are shown in Table 3, Figures 7 and 8, and Appendix C.

**Table 3.** Cluster centers of physical environmental characteristics of older neighborhoods (variables with * are standardized z-scores).

| Cluster Number | Clusters | | | | | | |
|---|---|---|---|---|---|---|---|
| | 1 | 2 | 3 | 4 | 5 | 6 | 7 |
| Parallel Index | 1 | 1 | 1 | 1 | 1 | 1 | 0 |
| Perimetric Index | 0 | 0 | 0 | 0 | 1 | 0 | 0 |
| Tower Index | 1 | 0 | 0 | 1 | 0 | 0 | 1 |
| Construction Time | 4 | 3 | 3 | 3 | 2 | 3 | 4 |
| Elevator Installation | 1 | 0 | 0 | 1 | 0 | 0 | 1 |
| Associated with Danwei | 0 | 0 | 0 | 0 | 1 | 1 | 0 |
| Property Right | 3 | 3 | 3 | 3 | 3 | 1 | 3 |
| FAR (Plot Ratio) * | 0.35 | −0.26 | −0.33 | −0.12 | −0.50 | −0.29 | 2.20 |
| AVE (Average Floor Number) * | 0.65 | −0.32 | −0.46 | 0.32 | −0.58 | −0.31 | 1.79 |
| GRIn (Green Ratio) * | 0.09 | −0.14 | −0.58 | 0.99 | −0.04 | −0.04 | −0.54 |
| LdIn (Land Use Mix Index) * | 0.34 | −1.57 | 0.48 | 0.26 | 0.18 | 0.17 | 0.04 |
| Property Fee * | 3.71 | −0.23 | −0.23 | −0.06 | −0.22 | −0.23 | 0.29 |
| Building Density * | −0.24 | 0.09 | 0.17 | −0.66 | 0.15 | 0.03 | 0.51 |

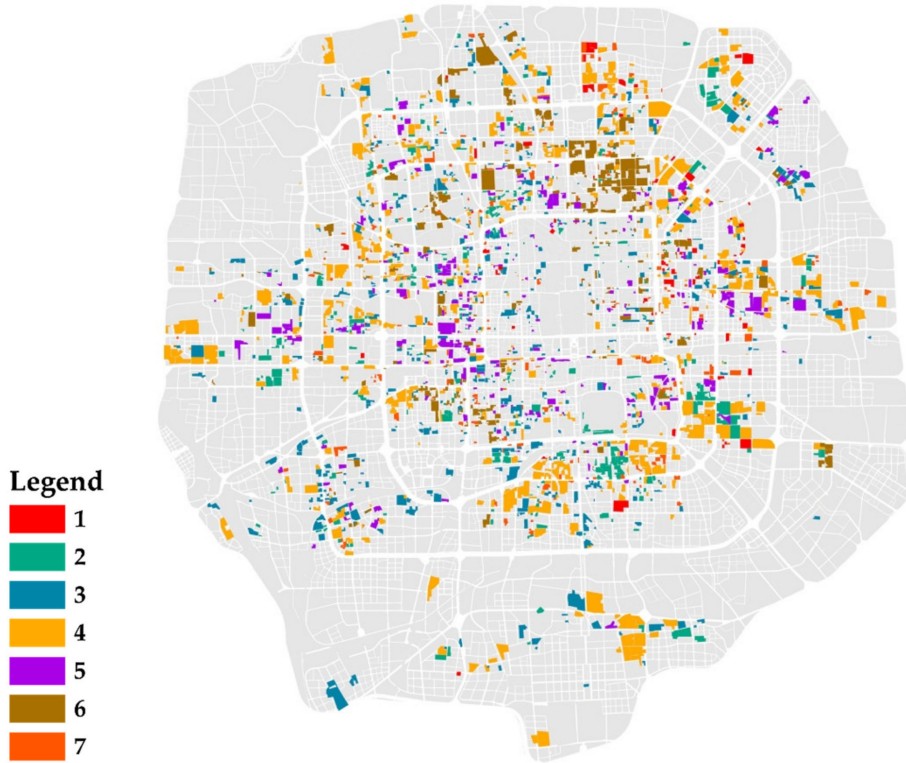

**Figure 7.** Classification Results of physical environmental characteristics of the selected older neighborhoods in Beijing.

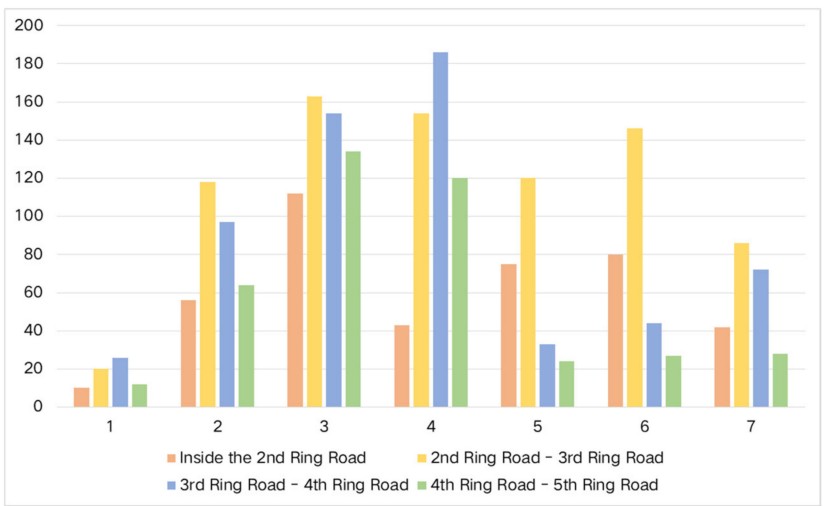

**Figure 8.** The spatial distribution of physical environmental classifications of the selected older neighborhoods.

After the ANOVA (Analysis of Variance) of classification (Table A6), this paper discovered that the clusters are significant for all physical environmental indexes ($p < 0.01$), which means that the seven clusters attained from classification analysis were significantly different from each other in the features of physical environmental indexes. Among the seven types of older neighborhoods obtained by K-means classification, the characteristics of type 2, 3, 5, and 6 old neighborhoods (named Group 1) were significantly different from type 1, 4 and 7 old neighborhoods (named Group 2). Most of the older neighborhoods in Group 1 used to be danwei communities and feature a "perimetric layout" or "parallel layout". Besides, the plot ratio and average floor number of buildings in Group 1 is relatively low and most of them are confronted with the problems of low green ratio, low property fee and the

lack of elevator installation. The differences among clusters within Group 1 are as follows: Cluster 5 which mostly located around the Second Ring Road has the earliest construction time, e.g., the Baiwanzhuang Neighborhood; Cluster 3 is a relatively scattered residential neighborhood with a high land use mix index and complete supporting service facilities, e.g., Huaibaishu Street North Block; Cluster 2 is mainly located between the second and third ring roads and was constructed on a large scale with a low land use mix index and limited supporting service facilities, e.g., Jinsong District 1. The older neighborhoods in Group 2 were mainly constructed after 1989 and most of them feature a high level of construction and management. Additionally, this kind of residential neighborhood is always equipped with towers and elevators. Besides, average floor numbers, plot ratio, green ratio and property fee in Group 2 have also improved. The differences among clusters in Group 2 are as follows: Cluster 1 has the highest physical environmental quality among the seven clusters, which features a "parallel layout" and "tower layout" and has a relatively higher green ratio, higher property fee and complete management and operation, e.g., the Bihu Residential Neighborhood; Cluster 4 is commercial housing with a relatively high green radio and property fee, which is located between third and fourth ring roads; Cluster 7 has the highest plot ratio and floor number, but its physical space quality is average with a relatively low green ratio.

The study conducted K-means classification on aging characteristics of selected older neighborhoods within the research scope and analyzes SSE when K is from 2 to 6, respectively (Figure 9).

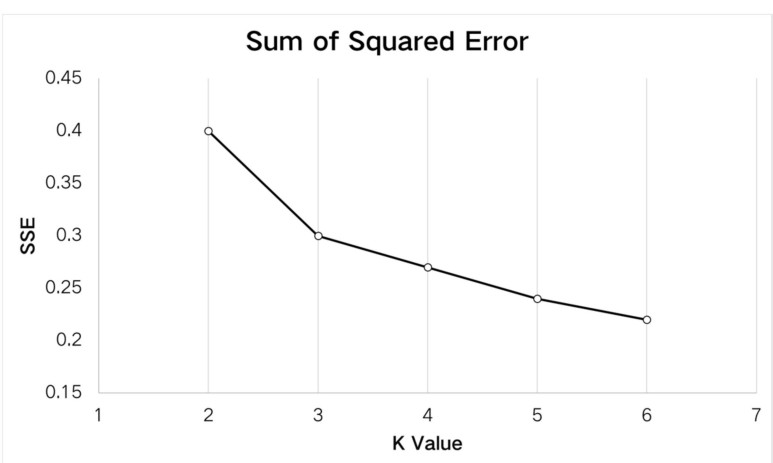

**Figure 9.** The SSE result analysis of K-means classification of the aging characteristics of the selected old neighborhoods in Beijing.

Therefore, this article determined the optimal number of clusters as three. This article conducted K-means classification according to three categories, and the results are shown in Table 4, Figure 10, Figure 11 and Appendix D.

**Table 4.** Cluster Centers of Aging Characteristics of Older Neighborhoods.

| Cluster Number | Clusters | | |
|---|---|---|---|
| | 1 | 2 | 3 |
| Proportion of Elderly People | 0.236 | 0.229 | 0.157 |
| Age: 65–70 | 0.330 | 0.268 | 0.396 |
| Age: 71–79 | 0.326 | 0.331 | 0.344 |
| Age: Above 80 | 0.344 | 0.401 | 0.260 |
| Education: High School or Below | 0.755 | 0.573 | 0.808 |
| Education: College or Above | 0.245 | 0.427 | 0.192 |
| Revenue Level: Below 3000 yuan | 0.394 | 0.280 | 0.595 |
| Revenue Level: Above 3000 yuan | 0.606 | 0.720 | 0.405 |
| Complete Self-Care Ability | 0.859 | 0.903 | 0.876 |
| Impaired Self-Care Ability | 0.123 | 0.080 | 0.107 |
| Inability of Self-Care | 0.018 | 0.017 | 0.017 |
| Marital Status: Unmarried | 0.914 | 0.859 | 0.914 |
| Marital Status: Married | 0.086 | 0.141 | 0.086 |
| Revenue Source: Pension Dependence | 0.952 | 0.952 | 0.895 |
| Revenue Source: Relief Dependence | 0.016 | 0.015 | 0.035 |
| Revenue Source: Family Support Dependence | 0.010 | 0.016 | 0.022 |

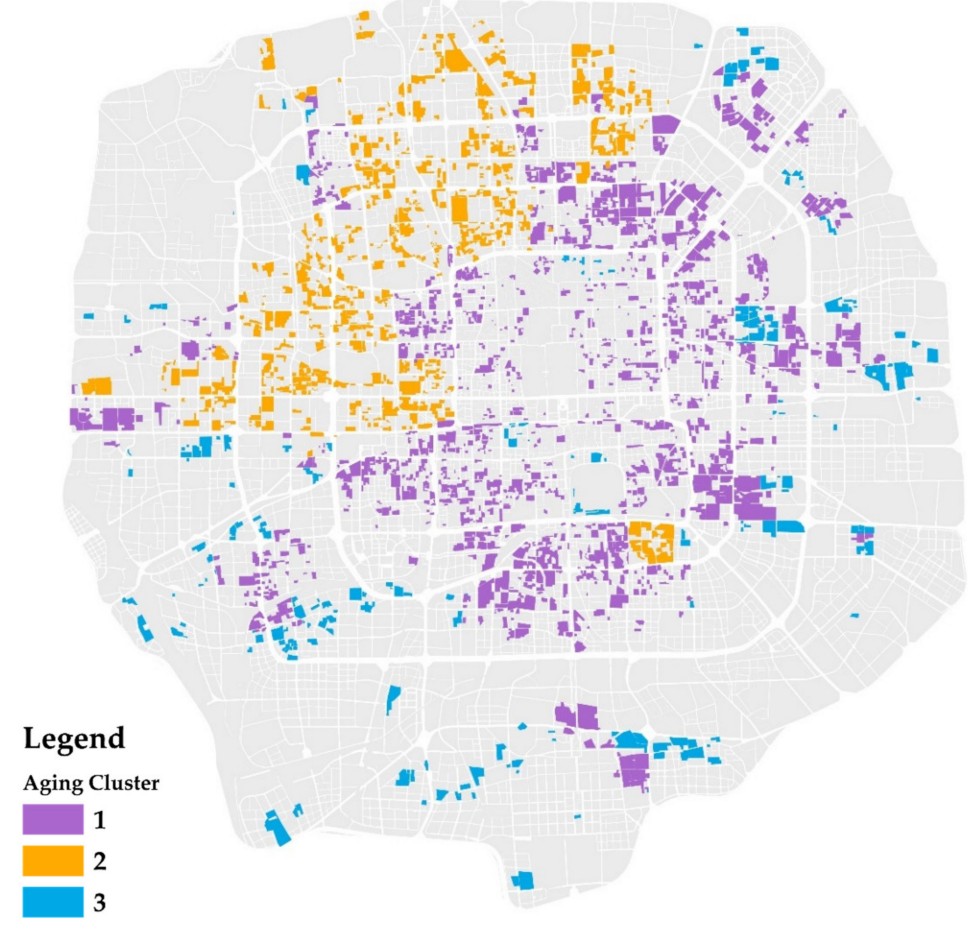

**Figure 10.** The classification result of the aging characteristics of the selected older neighborhoods in Beijing.

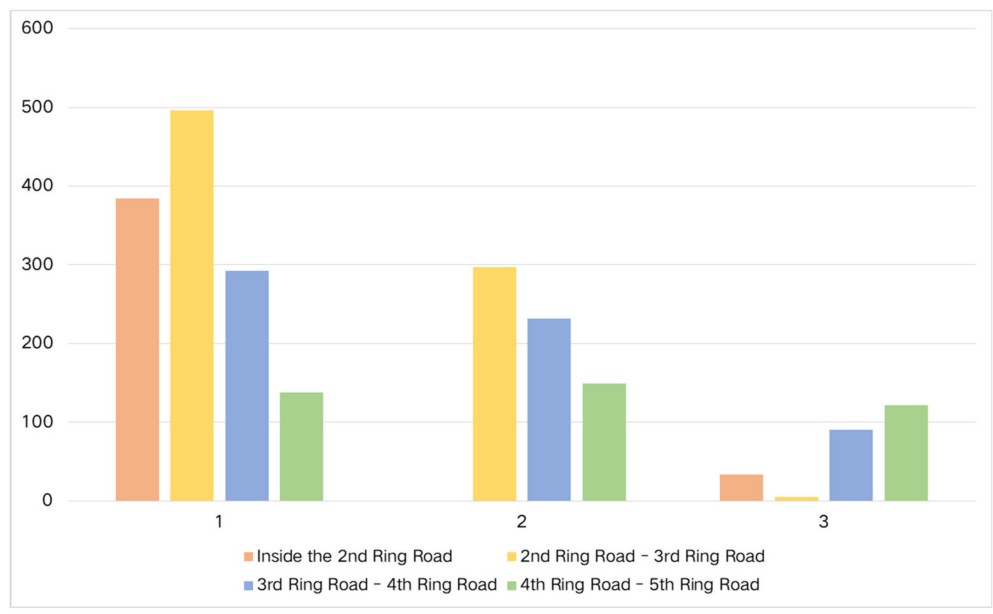

**Figure 11.** Spatial distribution of aging classifications of the selected older neighborhoods.

After the ANOVA (Analysis of Variance) of classification (Table A7), this paper discovered that the clusters have statistically significant differences for all aging indexes ($p < 0.01$), which means that the three clusters attained from the classification analysis are significantly different from each other in the features of aging indexes. There were significant differences among the three types of older neighborhood obtained by K-means classification based on aging characteristics: the Cluster 1 neighborhood and Cluster 2 neighborhood had a relatively higher proportion of the aging population and therefore the problems of aging were more serious. However, it is noteworthy that the proportion of the aging population and the elderly people with impaired self-care ability was relatively high and therefore the aging problems were more severe. This type of older neighborhood is mainly located in the core area of the capital function (Dongcheng District and Xicheng District) and the southeast of Beijing (Xingfu South Block, Xingfu North Block, and Xizhaosi West Block are the typical cases). However, the percentage of the oldest old (the aging population above 80) and the aging population with high education and high income of Cluster 2 was relatively high, meaning that it represents the higher social and economic status of the aged. This type of older neighborhood is mainly located in the northwest of the city, or areas along the west of Changan Avenue or Fangzhuang district in the southeast of the city, such as Shuangyushu East Block, Tayuan Community, and Fangqunyuan. The characteristics of Cluster 3 older neighborhood are the opposite from those of Cluster 2. Although the proportion of the aged or the oldest old (the aged above 80) in Cluster 3 was relatively low, the social and economic status of the aged living in this kind of residential neighborhood was relatively low. The majority of Cluster 3 is located in the periphery areas of the city, such as Fatou West Block.

*4.3. The Results of Interrelationship Analysis of Physical Environmental Classification and Aging Classification*

This article conducted an overlay analysis of the classification of the physical spatial characteristics and aging characteristics and the results are shown in Table 5 and Figures 12–14.

**Table 5.** The overlay analysis of the classification of the physical spatial characteristics and aging characteristics of the selected older neighborhoods.

| Number of Older Neighborhoods | | Physical Environmental Classification | | | | | | | Sum |
|---|---|---|---|---|---|---|---|---|---|
| | | 1 | 2 | 3 | 4 | 5 | 6 | 7 | |
| Aging Classification | 1 | 44 | 185 | 318 | 261 | 165 | 209 | 136 | 1318 |
| | 2 | 16 | 112 | 170 | 168 | 70 | 69 | 73 | 678 |
| | 3 | 8 | 38 | 75 | 74 | 17 | 19 | 19 | 250 |
| Sum | | 68 | 335 | 563 | 503 | 252 | 297 | 228 | 2246 |

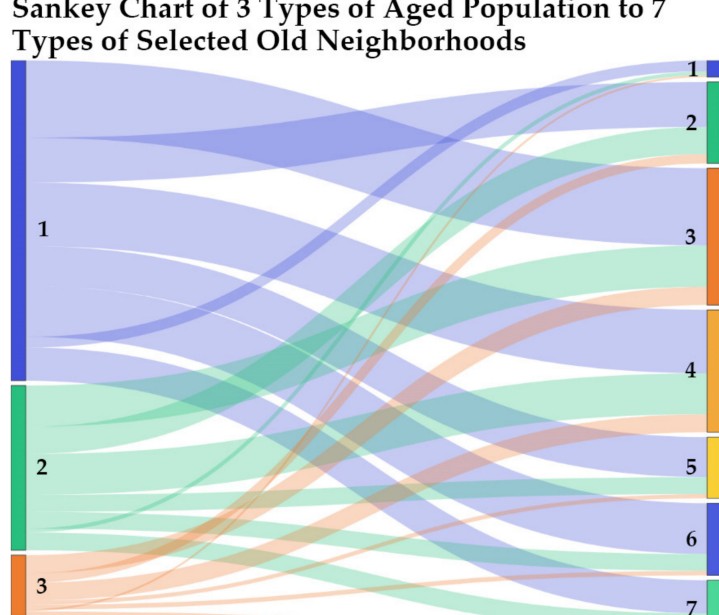

**Figure 12.** Sankey chart of the three types of aged population to the seven types of selected older neighborhoods.

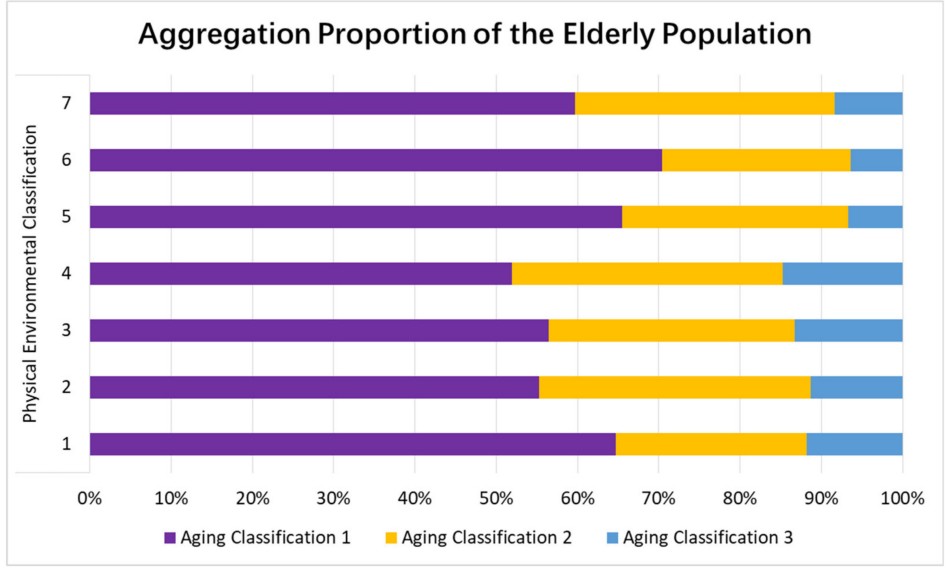

**Figure 13.** The aggregation proportion of the elderly population in the seven physical environmental clusters of the selected older neighborhoods.

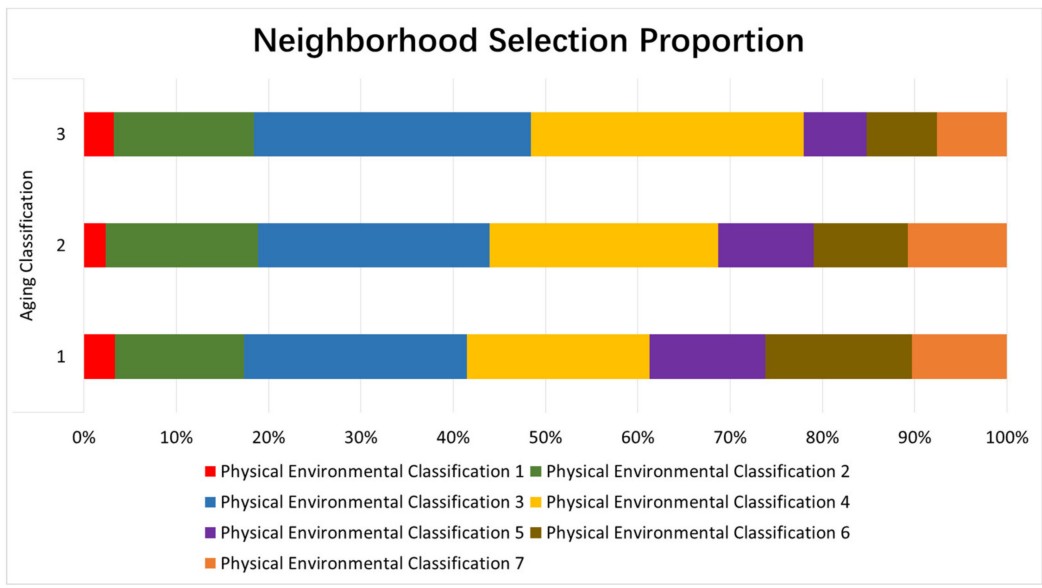

**Figure 14.** The neighborhood selection proportion of the three aging clusters of the selected older neighborhoods.

First, this research observed the aggregation characteristics of aged people in selected older neighborhoods and discovered that there was a certain similarity in the proportion of different kinds of aged populations among neighborhoods with similar physical environmental features. The selected older neighborhoods of Cluster 5 and Cluster 6 had similar population compositions. Compared with other clusters, the proportions of the aged population in Cluster 1 and Cluster 2 were relatively high in these neighborhoods. The selected older neighborhoods of Cluster 2, Cluster 3 and Cluster 4 had similar population composition. Compared with other clusters, the proportion of the aged population of Cluster 2 and Cluster 3 were relatively higher in these neighborhoods. The aging degree in the selected neighborhoods of Cluster 1 and Cluster 7 was between that of neighborhoods of the two groups mentioned above. The features mentioned above reveal that in comparison to other clusters of selected older neighborhoods, the proportion of the elderly with venerable age and low social status was relatively higher in the neighborhoods associated with the danwei community, which were mainly constructed before 1987. Additionally, compared to other clusters of selected neighborhoods, the proportion of the elderly with higher social status or low age was relatively high in the neighborhoods with the high level of construction and operation that were not associated with the danwei community.

On the other hand, in the perspective of selection preference of different kinds of the aged population, the research revealed that, compared with other kinds of aged population, the elderly of Cluster 3 had the propensity to live in the selected neighborhoods of Cluster 3 and Cluster 4, while the elderly of Cluster 1 had the propensity to live in the selected neighborhoods of Cluster 5 and Cluster 6. The selection preference of the aged population of Cluster 2 was between the above two clusters. Additionally, the aged population of Cluster 2 also had the propensity to live in the selected older neighborhoods of Cluster 2. The features above reveal that the aged population with venerable age and low social status are inclined to live in early-constructed neighborhoods that are associated with the danwei community, while the aged population with higher social income or lower age have the propensity to live in neighborhoods that were constructed later and are not associated with the danwei community.

## 5. Conclusions

According to the results, the spatial distribution of selected older neighborhoods in Beijing presents a distinct structure of single-central circling, expanding through the expansion process of the urban areas in Beijing. From the perspective of physical environment, the majority of the selected

older neighborhoods in Beijing are confronted with problems such as the lack of public space and supporting facilities and the inferiority of management and operation. Most of the neighborhoods that suffer from severe problems of the inferiority of physical environment are associated with danwei communities and located from the Second Ring Road to the Third Ring Road. In the perspective of aging, the proportion of both the aged population and the aged population with venerable age is relatively high in the selected older neighborhoods in Beijing and the aging problem is severe. Most of the older neighborhoods that suffer from severe problems of aging are in the northwest of Beijing, where there used to be plenty of danwei communities, or within the Third Ring Road.

Through the analysis of the classification results, this research has discovered that there is a certain regularity both in the composition of the aged in the selected older neighborhoods and the selection preference of the aged for the older neighborhoods. The elderly of venerable age and low social status are relatively inclined to live in early-constructed older neighborhoods that are associated with the danwei community. This type of older neighborhood is mainly concentrated in neighborhoods located between the second and the third ring roads, which were constructed earliest and are of relatively lower quality. Therefore, this type of older neighborhood is confronted with more severe problems in both the deterioration of physical environment and the aging of residence. Compared with the former, the housing choices of the elderly population of lower age or higher social status are relatively more diverse. For example, the elderly of lower age or higher social status are more likely to live in older neighborhoods constructed later by the private sector, of which the quality of construction and operation are relatively better than other older neighborhoods. In conclusion, this research has summarized both the physical environmental feature and the aging feature of selected older neighborhoods in Beijing through the analysis of multi-source data such as data from websites and can, to a certain extent, identify the relationship between the physical environmental features and the aging features of the selected older neighborhoods.

On the basis of the analysis of the physical environmental characteristics and aging characteristics of the selected old neighborhoods, this paper reflects on the existing policies and guidelines of the older neighborhoods in China and proposes some recommendations for future retrofitting policies. Firstly, the focus on retrofitting the old neighborhoods is not clear enough in practice at present. There are still some old neighborhoods with severe problems both in physical environment and in aging that have not been included in the renovation area. Therefore, the retrofitting guidelines based on the investigation and analysis of the older neighborhoods associated with danwei communities should be strengthened. Secondly, the existing retrofitting methods of older neighborhoods in China are too general and lack refined design guidelines based on physical environmental features and demographic conditions. Thus, the government should further work on the precise categorization and classification of different old neighborhoods to determine matched retrofitting elements. Thirdly, the Technical Standard for Comprehensive Renovation of Old Urban Residential Area released by the Chinese government in 2019 states that the needs of children, the elderly, the disabled and other special groups should be fully considered [64], while the current practices have not solved the needs of providing basic functions [65]. The renovation of the older neighborhoods should take aging conditions into consideration in order to satisfy the needs of daily life through elevator installation, providing sufficient public space for activity, and maintaining public facilities and the pavement of footpaths, etc.

## 6. Discussion

The older neighborhoods in Chinese cities are characterized by the concentration of the elderly. The community and people have undergone the processes of "co-growth" and "co-aging". In this context, the continuation of the lifespan of older people means that older neighborhoods need to promote the health and well-being of older adults. This requires environmental retrofitting with the goal of creating an aging-friendly community. From the perspective of aging, this paper addressed the lack of universality in the study of retrofitting older neighborhoods. Through qualitative and quantitative analysis in selected older neighborhoods in Beijing, the characteristics of the aging

physical environment and aging population of older neighborhoods were measured and described, supplementing the detailed features of the "double aging" phenomenon which has received no further analysis in existing research [66,67]. Besides, compared with other residential types, retrofitting the older neighborhoods to improve elderly people's quality of life is more urgent, but the relevant results of the findings are about housing-type design and barrier-free design in apartments or nursing institutions for the aged [68]. It is beneficial to create an aging-friendly community by selecting older neighborhoods as research objects to analyze the aging of Beijing's population.

To sum up, the study may have two potential implications. Firstly, two sets of measurement systems constructed on the Beijing case study can be referenced by cities that also have the feature of "synchronous aging", which can provide a scientific basis to improve the effectiveness of large-scale community retrofitting. In addition, the macro-scale analysis of the "co-aging" characteristics of the selected older neighborhoods was conducive to obtaining accurate screening of cases when conducting empirical research at the meso-micro scale. It is of great significance to carry out more detailed environmental retrofitting studies in older neighborhoods which have poor physical space quality and a high concentration of elderly people with lower socioeconomic status. However, this research also needs to be further improved. On the one hand, to the physical environment measurement of the older neighborhoods can be added the measurement of the internal facilities and outdoor public space, which is suitable for providing more practical suggestions. On the other hand, the community environment of the older neighborhoods includes two parts: the physical and social infrastructure. Future studies can investigate the social interaction and retrofitting needs in different types of older neighborhoods on the basis of the founding at the macro scale, making environmental retrofitting more in line with the goals of an aging-friendly community.

**Author Contributions:** Conceptualization, M.C. and C.G.; methodology, C.G. and M.C.; software, C.G.; validation, C.G., M.C.; formal analysis, M.C. and C.G.; investigation, M.C. and C.G.; resources, M.C. and C.G.; data curation, M.C. and C.G.; writing—Original draft preparation, M.C. and C.G.; writing—review and editing, M.C., C.G. and P.R.; visualization, M.C. and C.G.; supervision, P.R.; project administration, M.C. and C.G.; All authors have read and agreed to the published version of the manuscript.

**Funding:** This research received no external funding.

**Acknowledgments:** Thank you to Ying Long and Jiayan Liu of the School of Architecture, Tsinghua University, who provided technical support, including data for the residential district in Beijing and the aging population in Beijing.

**Conflicts of Interest:** The authors declare no conflict of interest.

## Appendix A

**Table A1.** Descriptive statistics of the physical environmental characteristics of old neighborhoods in Beijing: (**a**) plot ratio; (**b**) building density; (**c**) average floor number; (**d**) green ratio; (**e**) land use mix index; (**f**) property fee.

| | Min | Max | Median | Standard Deviation |
|---|---|---|---|---|
| (a) FAR (Plot Ratio) | 0.254 | 12.371 | 2.005 | 1.204 |
| (b) DEN (Building Density) | 0.047 | 0.798 | 0.321 | 0.092 |
| (c) AVE (Average Floor Number) | 1 | 30 | 5.682 | 3.274 |
| (d) GRIn (Green Ratio) | 0 | 0.673 | 0.14 | 0.112 |
| (e) LDIn (Land Use Mix Ratio) | 0.046 | 0.829 | 0.567 | 0.123 |
| (f) Property Fee (Yuan) | 0.100 | 9.800 | 1.398 | 1.118 |

**Table A2.** Descriptive statistics of the physical environmental characteristics of old neighborhoods in Beijing: construction time.

|  | N | Percentage |
|---|---|---|
| Construction Time: 1949–1957 | 109 | 4.9% |
| Construction Time: 1958–1976 | 297 | 13.2% |
| Construction Time: 1977–1988 | 940 | 41.9% |
| Construction Time: 1989–1999 | 900 | 40.1% |

**Table A3.** Descriptive statistics of the physical environmental characteristics of old neighborhoods in Beijing: Property Right Index.

|  | N | Percentage |
|---|---|---|
| Property Right: Unit | 349 | 15.5% |
| Property Right: Government | 32 | 1.4% |
| Property Right: Private Sector | 1865 | 83.0% |

**Table A4.** Descriptive statistics of the physical environmental characteristics of old neighborhoods in Beijing: (**a**) associated with danwei; (**b**) parallel index; (**c**) perimetric index; (**d**) tower index (**e**) elevator installation.

|  | 0 | | 1 | |
|---|---|---|---|---|
|  | N | Percentage | N | Percentage |
| (a) Associated with Danwei | 1136 | 50.6% | 1110 | 49.4% |
| (b) Parallel Index | 435 | 19.4% | 1811 | 80.6% |
| (c) Perimetric Index | 1472 | 65.5% | 774 | 34.5% |
| (d) Tower Index | 1506 | 67.1% | 740 | 32.9% |
| (e) Elevator Installation | 1297 | 57.7% | 949 | 42.3% |

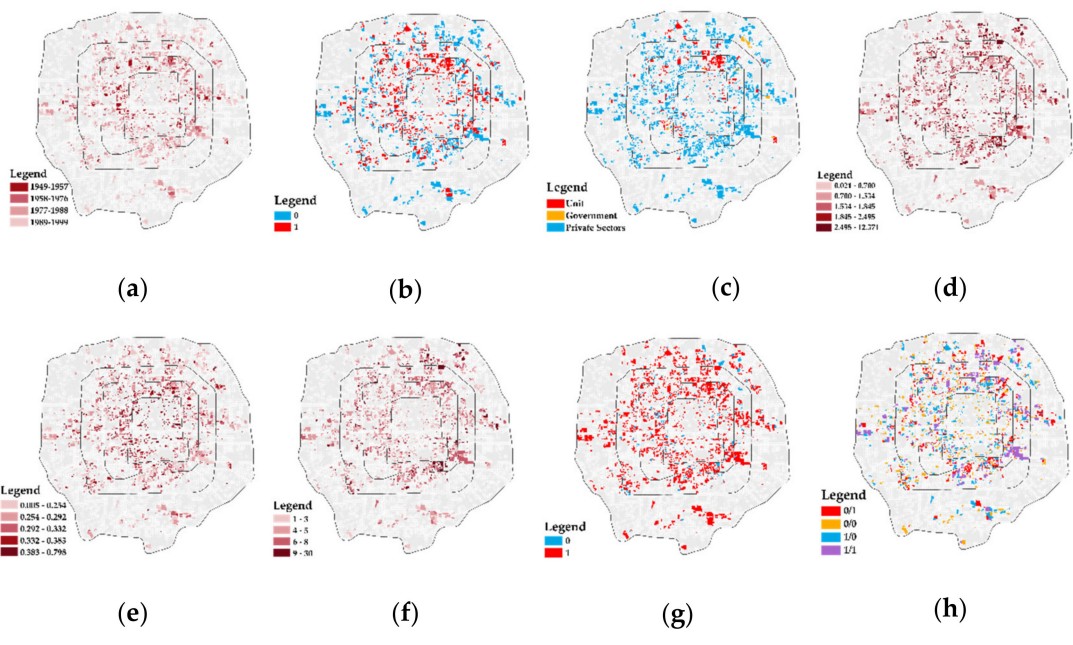

**Figure A1.** *Cont.*

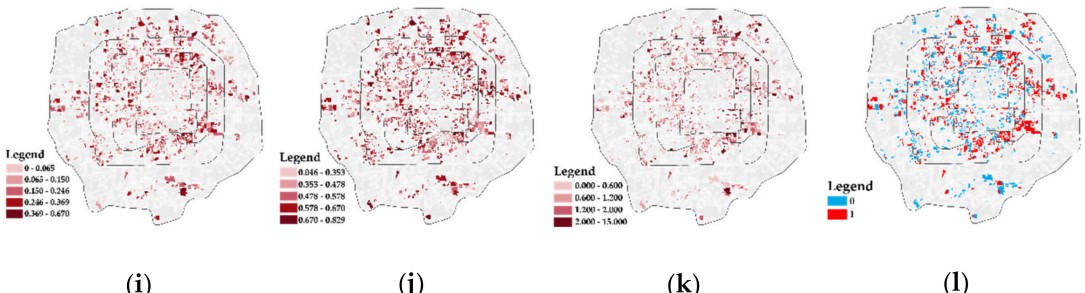

(i)         (j)         (k)         (l)

**Figure A1.** Spatial distribution of the physical spatial characteristics of selected old neighborhoods: (**a**) Construction time; (**b**) associated with danwei community; (**c**) property right index; (**d**) plot ratio; (**e**) building density; (**f**) average floor number; (**g**) parallel index; (**h**) perimetric/tower index; (**i**) green ratio; (**j**) land use mix index; (**k**) property fee; (**l**) elevator installation.

## Appendix B

**Table A5.** Descriptive Statistics of the ageing characteristics of old neighborhoods in Beijing.

|  | Min | Max | Mean | Standard Deviation |
|---|---|---|---|---|
| Proportion of Elderly People | 0.054 | 0.361 | 0.225 | 0.057 |
| Age: 65–70 | 0.151 | 0.475 | 0.319 | 0.058 |
| Age: 71–79 | 0.252 | 0.452 | 0.329 | 0.032 |
| Age: Above 80 | 0.156 | 0.568 | 0.352 | 0.069 |
| Education: High School or Below | 0.331 | 0.985 | 0.706 | 0.105 |
| Education: College or Above | 0.015 | 0.669 | 0.294 | 0.105 |
| Revenue Level: Below 3000 | 0.156 | 0.956 | 0.382 | 0.111 |
| Revenue Level: Above 3000 | 0.044 | 0.844 | 0.618 | 0.111 |
| Complete Self-Care Ability | 0.672 | 0.960 | 0.874 | 0.050 |
| Impaired Self-Care Ability | 0.033 | 0.320 | 0.108 | 0.049 |
| Inability of Self-Care | 0.006 | 0.081 | 0.018 | 0.007 |
| Marital Status: Unmarried | 0.774 | 0.945 | 0.897 | 0.039 |
| Marital Status: Married | 0.055 | 0.226 | 0.103 | 0.039 |
| Revenue Source: Pension Dependence | 0.460 | 0.990 | 0.946 | 0.037 |
| Revenue Source: Relief Dependence | 0.006 | 0.512 | 0.018 | 0.020 |
| Revenue Source: Family Support Dependence | 0.001 | 0.201 | 0.013 | 0.010 |

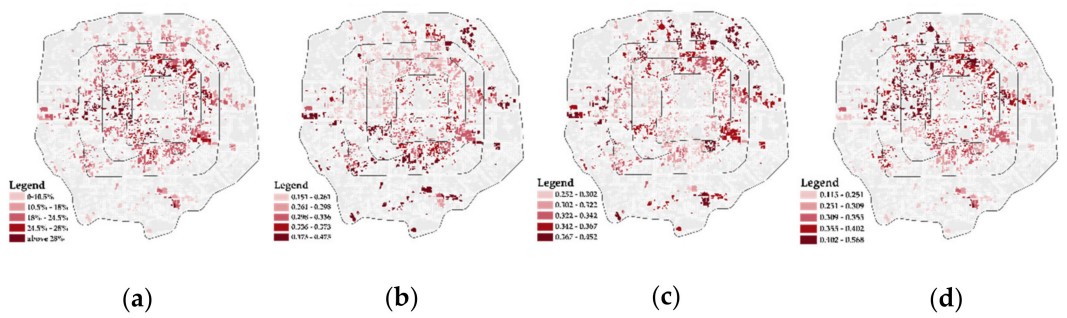

(a)         (b)         (c)         (d)

**Figure A2.** *Cont.*

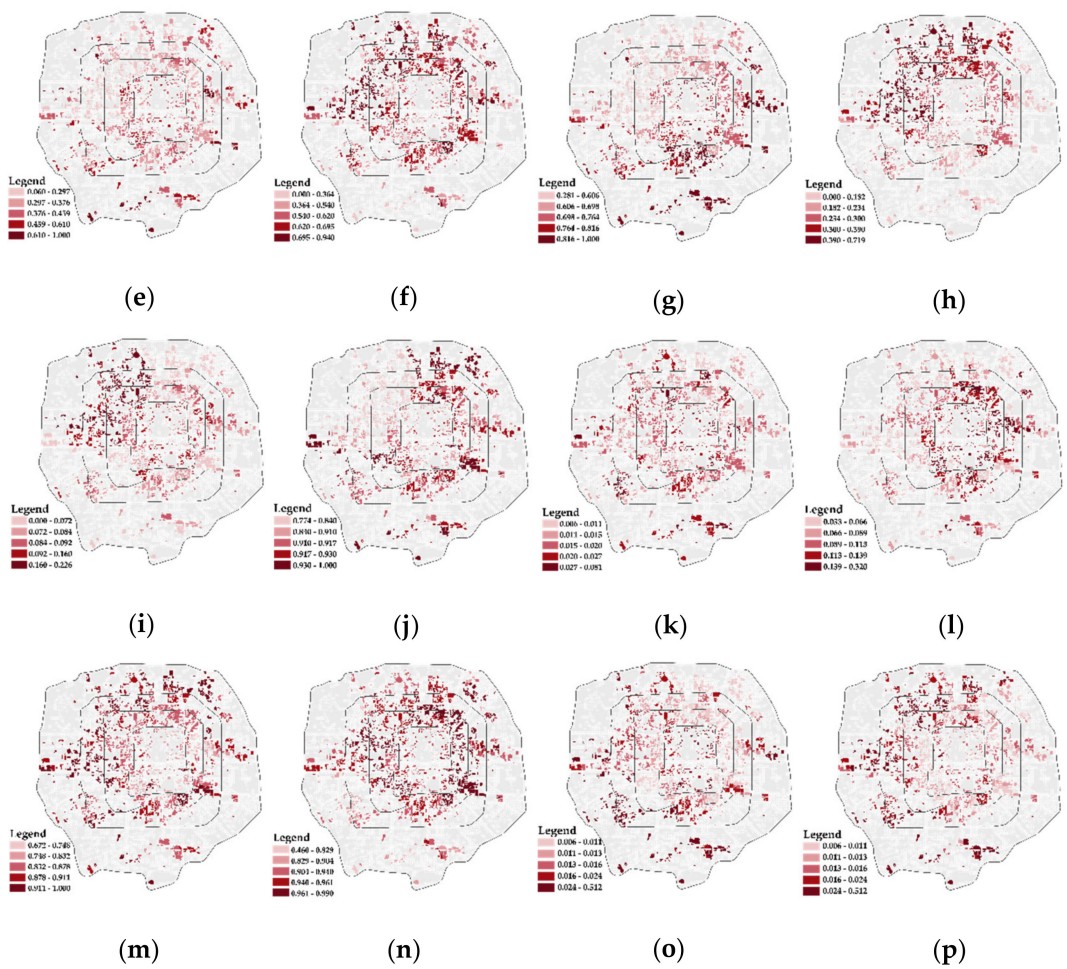

**Figure A2.** Spatial distribution of the aging characteristics of selected old neighborhoods: (**a**) proportion of elderly people; (**b**) 65–70; (**c**) 71–79; (**d**) above 80; (**e**) revenue below 3000; (**f**) revenue above 3000; (**g**) high school of below; (**h**) college or above; (**i**) unmarried; (**j**) married; (**k**) inability of self-care; (**l**) impaired self-care ability; (**m**) complete self-care ability; (**n**) pension dependence; (**o**) relief dependence; (**p**) family support dependence.

## Appendix C

**Table A6.** ANOVA analysis of the classification results of the physical environmental features.

|  | Cluster | | Error | | F | Significance |
|---|---|---|---|---|---|---|
|  | Mean Square | df | Mean Square | df |  |  |
| Parallel Index | 10.004 | 6 | 0.130 | 2239 | 77.049 | 0.000 |
| Perimetric Index | 3.191 | 6 | 0.218 | 2239 | 14.639 | 0.000 |
| Tower Index | 19.839 | 6 | 0.168 | 2239 | 117.776 | 0.000 |
| Construction Time | 124.800 | 6 | 0.365 | 2239 | 341.510 | 0.000 |
| Elevator Installation | 20.694 | 6 | 0.189 | 2239 | 109.312 | 0.000 |
| Associated with Danwei | 30.555 | 6 | 0.169 | 2239 | 180.942 | 0.000 |
| Property Right | 161.644 | 6 | 0.100 | 2239 | 1614.489 | 0.000 |
| FAR (Plot Ratio) | 214.556 | 6 | 0.386 | 2239 | 556.465 | 0.000 |
| AVE (Average Floor Number) | 179.559 | 6 | 0.524 | 2239 | 342.588 | 0.000 |
| GRIn (Green Ratio) | 126.640 | 6 | 0.636 | 2239 | 199.079 | 0.000 |
| LdIn (Land Use Mix Index) | 167.382 | 6 | 0.555 | 2239 | 301.803 | 0.000 |
| Property Fee | 171.507 | 6 | 0.405 | 2239 | 423.270 | 0.000 |
| Building Density | 51.052 | 6 | 0.593 | 2239 | 86.075 | 0.000 |

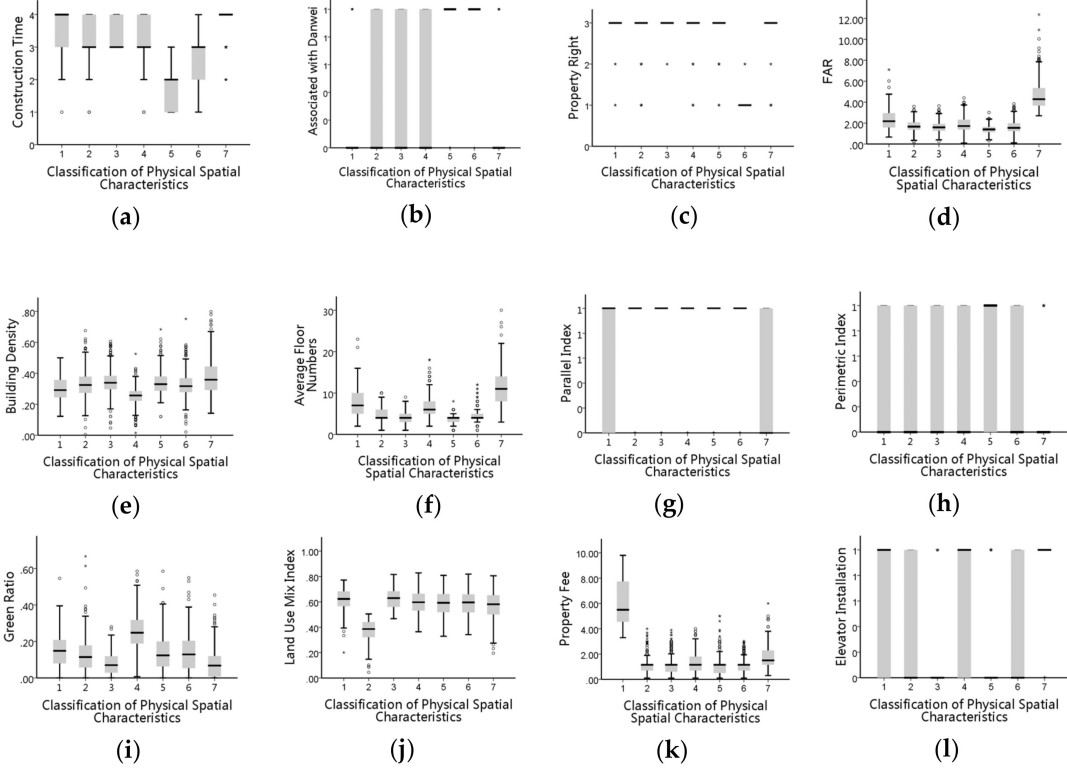

**Figure A3.** Index features of the classification of the physical spatial characteristics of selected old neighborhoods in Beijing: (**a**) construction time; (**b**) associated with danwei; (**c**) property right index; (**d**) plot ratio; (**e**) building density; (**f**) average floor number; (**g**) parallel index; (**h**) perimetric/tower index; (**i**) green ration; (**j**) land use mix index; (**k**) property fee; (**l**) elevator installation.

**Appendix D**

**Table A7.** ANOVA analysis of the classification results of aging features.

| | Cluster | | Error | | | |
| --- | --- | --- | --- | --- | --- | --- |
| | Mean Square | df | Mean Square | df | F | Significance |
| Proportion of Elderly People | 0.672 | 2 | 0.003 | 2243 | 257.232 | 0.000 |
| Age: 65–70 | 1.722 | 2 | 0.002 | 2243 | 930.364 | 0.000 |
| Age: 71–79 | 0.037 | 2 | 0.001 | 2243 | 37.647 | 0.000 |
| Age: Above 80 | 1.920 | 2 | 0.003 | 2243 | 619.735 | 0.000 |
| Education: High School or Below | 8.910 | 2 | 0.003 | 2243 | 2861.875 | 0.000 |
| Education: College or Above | 8.910 | 2 | 0.003 | 2243 | 2861.875 | 0.000 |
| Revenue Level: Below 3000 yuan | 9.300 | 2 | 0.004 | 2243 | 2355.685 | 0.000 |
| Revenue Level: Above 3000 yuan | 9.300 | 2 | 0.004 | 2243 | 2365.685 | 0.000 |
| Complete Self-Care Ability | 0.449 | 2 | 0.002 | 2243 | 218.441 | 0.000 |
| Impaired Self-Care Ability | 0.431 | 2 | 0.002 | 2243 | 216.615 | 0.000 |
| Inability of Self-Care | 0.000 | 2 | 0.000 | 2243 | 8.774 | 0.000 |
| Marital Status: Unmarried | 0.705 | 2 | 0.001 | 2243 | 779.181 | 0.000 |
| Marital Status: Married | 0.705 | 2 | 0.001 | 2243 | 779.181 | 0.000 |
| Revenue Source: Pension Dependence | 0.367 | 2 | 0.001 | 2243 | 361.282 | 0.000 |
| Revenue Source: Relief Dependence | 0.043 | 2 | 0.000 | 2243 | 122.936 | 0.000 |
| Revenue Source: Family Support Dependences | 0.016 | 2 | 0.000 | 2243 | 190.018 | 0.000 |

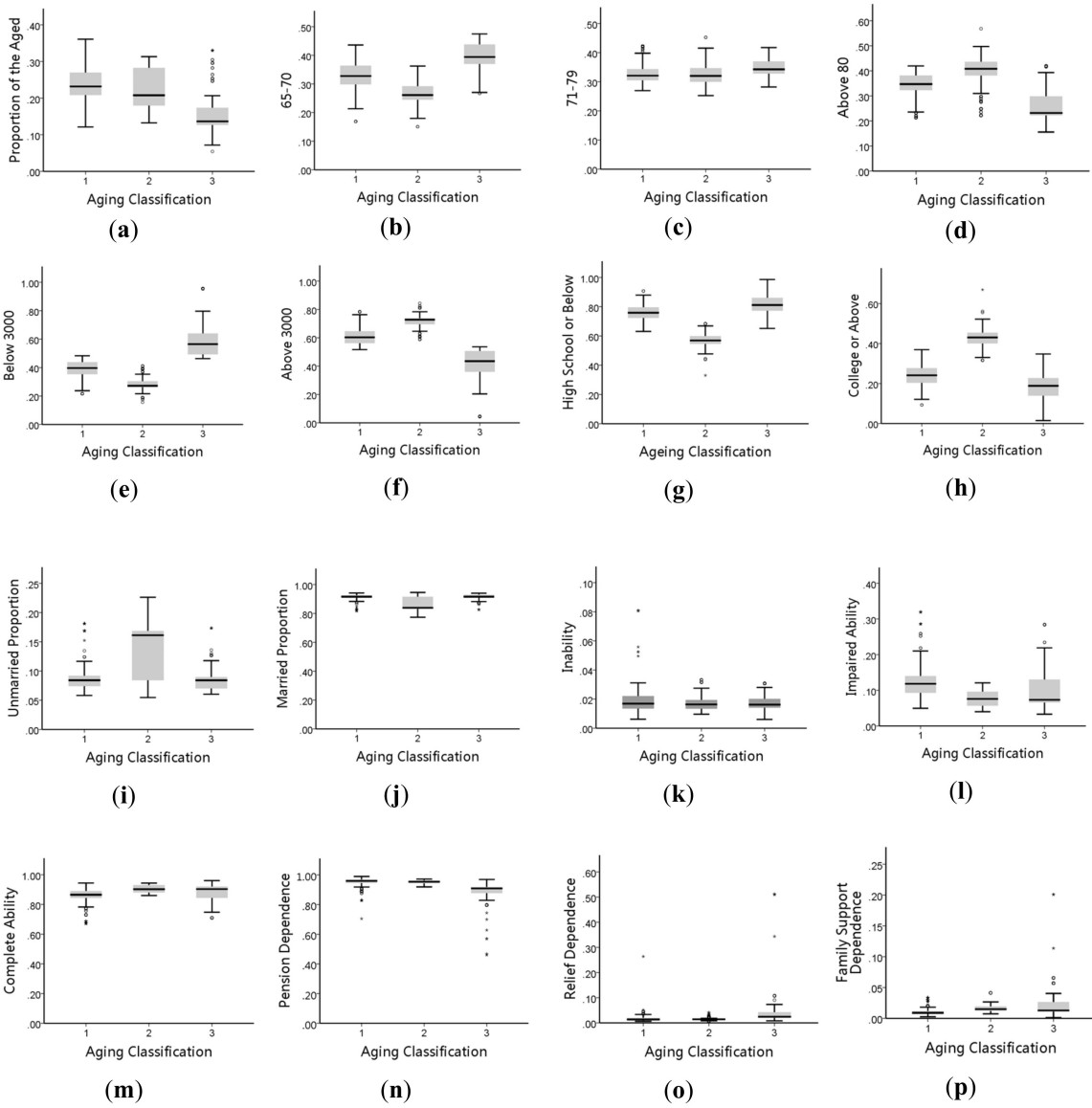

**Figure A4.** Index features of classification of the aging characteristics of selected old neighborhoods in Beijing: (**a**) proportion of elderly people; (**b**) 65–70; (**c**) 71–79; (**d**) above 80; (**e**) revenue below 3000; (**f**) revenue above 3000; (**g**) high school of below; (**h**) college or above; (**i**) unmarried; (**j**) married; (**k**) inability of self-care; (**l**) impaired self-care ability; (**m**) complete self-care ability; (**n**) pension dependence; (**o**)relief dependence; (**p**) family support dependence.

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
