# Peer review of "Beijing’s Selected Older Neighborhoods Measurement from the Perspective of Aging"

_sustainability, doi:10.3390/su12104112_

Round 1
Reviewer 1 Report
The authors have addressed my previous concerns. A final proofreading will be beneficial, otherwise i have no other further comments.
Author Response
Dear Reviewer:
Thank you for your comment concerning our manuscript entitled” Beijing Selected Older Neighborhoods Measurement from the Perspective of Ageing” with a previous ID sustainability-759503. We have finished the proof reading and checking carefully, and some corrections about the proof are provided below.
Corrections:
- In paper topic, the “older neighborhood” should be plural version and changed into“older neighborhoods”. Many same plural mistakes are corrected in this revision.
- In line 16, the “and so on” should be”, etc.,”.
- In line 32, “meso and micro scale” should be “middle-scale and micro-scale”.
- In line 36, remove the “most” in front of the “essential”.
- In line 62, the “above-mentioned phenomenon” should be phenomenon mentioned above.
- In line 110, “it not only encompasses” should be “it encompasses not only”.
- In line 176, “Beijing provide” should be “Beijing provides”.
- In line 305, the sentence on is not complete. It should be corrected by deleting extra words, “and the”.
- In line 537, “in this kind or” should be “in this kind of”.
- In line 569, “other cluster” should be “other clusters”.
- In line 616, “to certain extent” should be “to a certain extent”.
We greatly appreciate your professional comment and hope that the corrections will meet with approval.

Reviewer 2 Report
The paper has improved by taking into account the reviewers' reports. However, there are still some issues that need to be addressed.
On lines from 156-166, the 5 design features that are claimed to be derived from the literature, are mentioned. However, these features were not introduced in the introduction section where the literature review was discussed. Therefore, it is still not clear how the mentioned physical characteristics of this study are determined.
The study presents the results of several analyses. However, these results are not summarized in the discussion and conclusions sections. Due to that, there is no synthesis of the findings. In addition, the discussion of results can give reference to the findings. The findings should be supported by existing literature.
The conclusion section does not give any policy suggestions based on findings. It is not clear how the findings of this study will be used by policymakers or urban planners. This should be articulated, especially referring back to the motivation and the main research aim of the study.
The quality of the figures is very low. Figures are very hard to read.
The sentence on line 235 is not complete. Throughout the text, there are spelling mistakes that should be checked.
Author Response
Dear Reviewer:
Thank you for your comments concerning our manuscript entitled” Beijing Selected Older Neighborhoods Measurement from the Perspective of Ageing” with a previous ID sustainability-759503. Those comments are all valuable and very helpful for revising and improving our paper, as well as the important guiding significance to our researches. We have studied comments carefully and have made corrections which we hope meet with approval. The response to the reviewer’s comments are as flowing”:
- Response to comment: On lines from 156-166, the 5 design features that are claimed to be derived from the literature are mentioned. However, these features were not introduced in the introduction section where the literature review was discussed.
Response: We have made corrections according to the comments. The literature review about 5 design features has been added in the last paragraph of the introduction part.
- Response to comment: There is no synthesis of the findings. In addition, the discussion of results can give reference to the findings. The findings should be supported by existing literature.
Response: Considering the Reviewer’s comment, this revision has supplemented the relevant existing literature to support the findings in the final discussion part, especially about double aging study and elderly-oriented design research.
- Response to comment: The conclusion section does not give any policy suggestions based on findings, especially referring back to the motivation and the main research aim of the study.
Response: Considering the Reviewer’s comment, policy suggestions have been added to the conclusion section. This paper recommends that the government should focus more on the renovation of old neighborhoods associated with the danwei community and the government should further work on meticulous design guidelines to determine the different elements of renovation according to different spatial characteristics and aging conditions.
- Response to comment: The quality of the figures is very low. Figures are very hard to read.
Response: We are very sorry for our negligence. Figures 1, 2, 12, A1, B1, C1 and D1 have been replaced with figures with better quality. To make other figures clearer, the legends of Figures 3, 4, 5, 7, 10, have been replaced.
- Response to comment: The sentence on line 235 is not complete. Throughout the text, there are spelling mistakes that should be checked.
Response: The grammar mistake in sentence has been corrected by deleting extra words “and the” on line 305. We also have corrected spelling mistakes after the proof reading.
Special thanks to you for your good comments and hope that the correction will meet with approval.

Reviewer 3 Report
The authors introduced proper changes to the paper. In my opinion the paper after rewriting presents the proper academic level and may be published. Some editing comments below:
- Figures 1, 2, C1 and D1 could have better quality
- Figure 7 and 8 titles – minor editing mistakes – spaces needed
Author Response
Dear Reviewer:
Thank you for your comments concerning our manuscript entitled” Beijing Selected Older Neighborhoods Measurement from the Perspective of Ageing” with a previous ID sustainability-759503. Those comments are all valuable and very helpful for revising and improving our paper, as well as the important guiding significance to our researches. We have studied comments carefully and have made correction which we hope meet with approval. The responds to the reviewer’s comments are as flowing”:
- Response to comment: Figures 1, 2, C1 and D1 could have better quality
Response: Considering the Reviewer’s comment, the figures mentioned have been replaced with figures with better quality.
- Response to comment: Figure 7 and 8 titles – minor editing mistakes – spaces needed.
Response: The spaces missed in Figure 7 and 8 titles have been added.
Special thanks to you for your good comments and hope that the correction will meet with approval.

Reviewer 4 Report
Reviewed article is very interesting. The study aims to examine assumes connections between older neighborhoods and ageing. Using multi-source data to identify relationships between the physical environmental features and the aging features is a good idea. The method shown in article is interesting and can be used to measure this kind of research, and the description is clear. The results herein support public policy proposals relevant to urban planning, environmental design and aging policies Future studies can investigate the social interaction and retrofitting needs in different types of older neighborhood on the basis of the founding at macro scale, making the environmental retrofitting more in line with the goals of aging friendly community. Some figures are in bad quality, that should be improved. Good luck with your paperAuthor Response
Dear Reviewer:
Thank you for your comments concerning our manuscript entitled” Beijing Selected Older Neighborhoods Measurement from the Perspective of Ageing” with a previous ID sustainability-759503. Those comments are all valuable and very helpful for revising and improving our paper, as well as the important guiding significance to our researches. We have studied comments carefully and have made correction which we hope meet with approval. The responds to the reviewer’s comments are as flowing”:
Response to comment: Some figures are in bad quality, that should be improved.
Response: Figures 1, 2, 12, A1, B1, C1 and D1 have been replaced with figures with better quality. To make other figures clearer, the legends of Figure 3, 4, 5, 7, 10, have been replaced.
Special thanks to you for your good comments and hope that the correction will meet with approval.

Round 2
Reviewer 2 Report
The authors have addressed my previous concerns. I have no other further comments.
This manuscript is a resubmission of an earlier submission. The following is a list of the peer review reports and author responses from that submission.
Round 1
Reviewer 1 Report
This is an interesting paper that explored the built environment and spatial distributions of old neighborhoods in Beijing. This is a very important topic. However, before any further consideration, I will raise the following questions.
First and foremost, the paper’s storyline should be straightforward and simple. However, the authors put too much background information and obscured the story. I understand that the authors did lots of work, yet you don’t need to put all of them in the paper. Some of them should definitely go to appendix or simply mention the results. At least 30% contents/ tables/figures should go to the appendix. A table summary for literature review will be good.Too much information about calculation, backgrounds and section 3, readers will get lost after this. Similar thing about section 3.2 residential spatial temporal distribution, no need to provide a full review about urban planning process in Beijing, at least simplify it.
Rearrange Figure 4, it is very hard to see these small figures. Maybe put them into one landscape layout page. This also goes to Figure 5, which is extremely hard to read as well. Rule of thumb, if authors had some hard time to read these figures then just don’t include them in the paper. Also for Figure 7, can you honestly tell the different bar values in figure 7 d,e,f. I seriously doubt it. Remove some of these figures. Table 6 and Figure 10-11 are important findings, most of the other colorful figures should be in the appendix. Proofreading is needed one of the keywords was wrong: Ageing to Aging Last but not the least how Beijing government can use some of your findings to improve BE in these old neighborhoods and improve the people’s living situations in these areas.Reviewer 2 Report
Paper is about providing insights on elderly and their living environments for the elderly-oriented renovation of old neighborhoods. By means of qualitative and quantitative analysis, the paper investigates the physical environmental characteristics of neighborhoods and the characteristics of elderly population living in these neighborhoods. The paper contributes to the literature with a case study. However, there are still some issues to be improved.
In the introduction, the paper points out a gap in the literature as such "there is a need for a macro level approach rather than a case study". However, this current study is also using a case area analysis. It should be explained more explicitly how the current case study differs from the existing studies. Moreover, introduction section should clearly explain the actual contribution of this paper to literature. Currently it is not comprehensible what methodological and theoretical contribution of the paper is. In addition, paper needs a more thorough literature review especially to justify the selection of physical environmental characteristics and also the considered indicators. In current form, selected physical environment characteristics and indicators seem subjective to the authors' knowledge.
Danwei community is first mentioned on line 124 at materials and methods section. Danwei community is not known by all readers and therefore needs to be explained earlier.
In Materials and Methods section, selected analysis techniques (i.e. K-means clustering) needs to be discussed in the beginning of the section as such why these are the most suitable methodologies for this research.
In Quantitative results section, starting at line 331, differences between clusters are explained. It would be better to see if there are statistically significant differences between clusters in terms of socio-demographics. In that sense, bivariate analysis could be used.
In Discussion section, line 388, an index system is suggested for providing more practical suggestions to improve old neighborhoods. In that sense, an explanation is required for what is needed to be done in order to further improve the research, especially to determine an index system.
Finally, the paper should also have a conclusion section where the findings and main contribution of the study is evaluated by authors. Currently, this is missing.
Reviewer 3 Report
Reviewed article could be interesting, but some parts makes me confused and should be strengthened.
Introduction.The title of the article assumes connections between older neighborhoods and ageing. But this connections are not clearly shown. What is the reason of comparing this two components? The aim of the article isn’t clear.
The introduction should be extended with theoretical background – this part should contain more about recent surveys. In the line 50. we have: “On this issue, Chinese scholars have done a lot of researches” without examples and discussion.
Materials and methods.The method shown in article is interesting and can be used to measure this kind of research, but the description is confusing and unclear.
It is not clear, what authors understand as older neighborhood – is it based only on age of construction? What about neighborhoods built early and then restored?
It is not explained, what is Danwei? For Chinese readers it can be obvious, but has to be explained for international readers.
The construction time is divided into 4 periods. Why four? Why are they uneven? Why they do not correspond to the official five-year plans, described in other part of the article?
What means ½ health care? I’m sure, Chinese readers know health care system in China, but international readers will have problem with that.
Characteristic of survey area should be at the beginning of the article – to show to the readers specificity of the area under study.
Some figures are in bad quality, especially fig. no. 4, 5, 8 and 11.
Conclusion.There are no part entitled “Conclusions”. Some of them are in “Discussion”, but are really general.
In discussion part authors should discuss their findings with other surveys, but there are lack of discussion, for example with other part of China.
The reader doesn’t know, does this process and situation special for Bejing?
Good luck with your paper
Reviewer 4 Report
The paper presents the results of research concerning neighborhood aspects from the perspective of the aging process. The paper is generally prepared on a good academic level. However, it has some flaws that should be corrected.
Introduction
The introduction is mostly restricted to Chinese issues. I think that some general outlines of problems described in the paper from international literature (not only from China) should also be cited and proper references are given.
Material and methods
As the paper is addressed to the international community not necessarily familiar to specific regulation and history of China, I suggest explaining the term ‘Danwei’.
The text in lines 153-157 should be simplified and shortened, i.e. if given feature appears then index equals 1, if not than 0.
It should be explained why the income revenue of 3000 yuan is selected. It should be good to add the ‘yuan’ unit in figure 2.
Qualitative Description
Units should be provided in table 2.
Quantitative Results
I do not think that the number of old neighborhoods should appear in the first column of tables 3 and 4.
Table 3 is not clear. If the described features, for example ‘Construction Time’ or ‘Property Fee’ have a unit, it should be given. I think that describing features having only values ‘0’ or ‘1’ with median and standard deviation does not make sense. It is better to give percentages of ones having some feature or not.
Table 6 – the sums of values of subsequent characteristics in particular clusters (age, degree of health care revenue) should give ‘1’ in total.
Table 7 – the sums of values in rows for aging classification clusters 1 and 2 is not 100%. It should be corrected.
In my opinion, the research results described in the following paragraph (lines 354-371) are not supported by the conducted study. I suggest adding an additional row and column for total amounts. It should help authors to analyze the research outcome.
Discussion
It looks rather like conclusions with some of the statements being rather obvious.
References
The references concerning Chinese issues seem to be sufficient. As indicated earlier, some more references concerning these aspects outside China should be added.
Editorial issues
'5th Year Plan' should be written in the same way in the whole text.
Some editorial mistakes i.a. lack of space should be corrected – lines 54, 58, 121, 132, 166, 200, 219, 227 and 261.